# Real-time monitoring of peptidoglycan synthesis by membrane-reconstituted penicillin-binding proteins

**Víctor M Hernández-Rocamora[1†], Natalia Baranova[2†‡], Katharina Peters[1§], Eefjan Breukink[3], Martin Loose[2*], Waldemar Vollmer[1*]**

[1]Centre for Bacterial Cell Biology, Biosciences Institute, Newcastle University, Newcastle upon Tyne, United Kingdom; [2]Institute for Science and Technology Austria (IST Austria), Klosterneuburg, Austria; [3]Membrane Biochemistry and Biophysics, Bijvoet Centre for Biomolecular Research, University of Utrecht, Utrecht, Netherlands

**\*For correspondence:**
mloose@ist.ac.at (ML);
w.vollmer@ncl.ac.uk (WV)

[†]These authors contributed equally to this work

**Present address:** [‡] Department of Pharmaceutical Chemistry, Faculty of Life Sciences, University of Vienna, Vienna, Austria; [§] School of Natural & Environmental Sciences, Faculty of Science, Agriculture & Engineering, Newcastle University, Newcastle upon Tyne, United Kingdom

**Competing interests:** The authors declare that no competing interests exist.

**Abstract** Peptidoglycan is an essential component of the bacterial cell envelope that surrounds the cytoplasmic membrane to protect the cell from osmotic lysis. Important antibiotics such as β-lactams and glycopeptides target peptidoglycan biosynthesis. Class A penicillin-binding proteins (PBPs) are bifunctional membrane-bound peptidoglycan synthases that polymerize glycan chains and connect adjacent stem peptides by transpeptidation. How these enzymes work in their physiological membrane environment is poorly understood. Here, we developed a novel Förster resonance energy transfer-based assay to follow in real time both reactions of class A PBPs reconstituted in liposomes or supported lipid bilayers and applied this assay with PBP1B homologues from *Escherichia coli, Pseudomonas aeruginosa,* and *Acinetobacter baumannii* in the presence or absence of their cognate lipoprotein activator. Our assay will allow unravelling the mechanisms of peptidoglycan synthesis in a lipid-bilayer environment and can be further developed to be used for high-throughput screening for new antimicrobials.

## Introduction

Peptidoglycan (PG) is a major cell wall polymer in bacteria. It is composed of glycan strands of alternating N-actetylglucosamine (Glc*N*Ac) and N-acetylmuramic acid (Mur*N*Ac) residues interconnected by short peptides. PG forms a continuous, mesh-like layer around the cell membrane to protect the cell from bursting due to the turgor and maintain cell shape (*Vollmer et al., 2008*). The essentiality and conservation of PG in bacteria make PG metabolism an ideal target of antibiotics.

Class A penicillin-binding proteins (PBPs) are bifunctional PG synthases which use the precursor lipid II to polymerize glycan chains (glycosyltransferase [GTase] reactions) and crosslink peptides from adjacent chains by DD-transpeptidation (*Goffin and Ghuysen, 1998*). Moenomycin inhibits the GTase and β-lactams the transpeptidase function of class A PBPs (*Sauvage and Terrak, 2016*; *Macheboeuf et al., 2006*). In *Escherichia coli*, PBP1A and PBP1B account for a substantial proportion of the total cellular PG synthesis activity (*Cho et al., 2016*) and are tightly regulated by interactions with multiple proteins (*Egan et al., 2015*; *Typas et al., 2012*; *Egan et al., 2020*; *Egan et al., 2017*), including the outer membrane-anchored activators LpoA and LpoB (*Egan et al., 2018*; *Typas et al., 2010*; *Jean et al., 2014*).

Historically, in vitro PG synthesis assays have been crucial to decipher the biochemical reactions involved in PG synthesis and determine the mode of action of antibiotics (*Izaki et al., 1968*). However, these studies were limited by the scarcity of lipid II substrate and the inability to purify a sufficient quantity of active enzymes. Lipid II can now be synthesized chemically (*VanNieuwenhze et al.,*

*2002*; *Schwartz et al., 2001*; *Ye et al., 2001*) or semi-enzymatically (*Breukink et al., 2003*; *Egan et al., 2015*), or isolated from cells with inactivated MurJ (*Qiao et al., 2017*). Radioactive or fluorescent versions of lipid II are also available to study PG synthesis in a test tube. However, there are several drawbacks with currently available PG synthesis assays. First, most assays are end-point assays that rely on discrete sampling and therefore do not provide real-time information about the enzymatic reaction. Second, some assays involve measuring the consumption of lipid II or analysing the reaction products by SDS-PAGE (*Egan et al., 2015*; *Barrett et al., 2007*; *Qiao et al., 2014*; *Sjodt et al., 2018*) or high-pressure liquid chromatography (HPLC) after digestion with a murami-dase (*Bertsche et al., 2005*; *Born et al., 2006*). These laborious techniques make assays incompatible with high-throughput screening and hinder the determination of kinetic parameters. A simple, real-time assay with dansyl-labelled lipid II substrate overcomes these problems but is limited to assay GTase reactions (*Schwartz et al., 2001*; *Offant et al., 2010*; *Egan et al., 2015*).

Recently, two types of real-time TPase assays have been described. The first uses non-natural mimics of TPase substrates such as the rotor-fluorogenic 470 D-lysine probe Rf470DL, which increases its fluorescence emission upon incorporation into PG (*Hsu et al., 2019*). The second assay monitors the release of D-Ala during transpeptidation in coupled enzymatic reactions with D-amino acid oxidase, peroxidases, and chromogenic or fluorogenic compounds (*Frére et al., 1976*; *Gutheil et al., 2000*; *Catherwood et al., 2020*). Coupled assays are often limited in the choice of the reaction conditions, which in this case must be compatible with D-amino acid oxidase activity. Hence, each of the current assays has its limitations and most assays exclusively report on either the GTase or TPase activity, but not both activities at the same time.

Another major drawback of many of the current assays is that they include detergents and/or high concentration (up to 30%) of the organic solvent dimethyl sulfoxide (DMSO) to maintain the PG synthases in solution (*Offant et al., 2010*; *Biboy et al., 2013*; *Huang et al., 2013*; *Lebar et al., 2013*; *Qiao et al., 2014*; *Egan et al., 2015*; *Catherwood et al., 2020*). However, both detergents and DMSO have been shown to affect the activity and interactions of *E. coli* PBP1B (*Egan and Vollmer, 2016*). Importantly, a freely diffusing, detergent-solubilized membrane enzyme has a very different environment compared to the situation in the cell membrane where it contacts phospholipids and is confined in two dimensions (*Gavutis et al., 2006*; *Zhdanov and Höök, 2015*). Here, we sought to overcome the main limitations of current PG synthesis assays and establish a system with more physiological experimental conditions. We used sensitive Förster resonance energy transfer (FRET) detection for simultaneous monitoring of GTase and TPase reactions. The real-time assay reports on PG synthesis in phospholipid vesicles or planar lipid bilayers. We successfully applied this assay to several class A PBPs from pathogenic Gram-negative bacteria, demonstrating its robustness and potential use in screening assays to identify PBP inhibitors.

## Results

### Real-time assay for detergent-solubilized *E. coli* PBP1B

To develop a FRET-based real-time assay for PG synthesis using fluorescently labelled lipid II, we prepared lysine-type lipid II versions with high quantum yield probes, Atto550 (as FRET donor) and Atto647n (as FRET acceptor), linked to position 3 (*Figure 1—figure supplement 1A, B*; *Mohammadi et al., 2014*; *Egan et al., 2015*). For assay development, we used *E. coli* PBP1B (PBP1B[Ec]) (*Egan et al., 2015*; *Bertsche et al., 2005*; *Biboy et al., 2013*) solubilized with Triton X-100 and a lipid-free version of its cognate outer membrane-anchored lipoprotein activator LpoB (*Typas et al., 2010*; *Egan et al., 2014*; *Egan et al., 2018*; *Lupoli et al., 2014*; *Catherwood et al., 2020*).

PBP1B[Ec] can utilize fluorescently labelled lipid II to polymerize long glycan chains only when unlabelled lipid II is also present in the reaction (*Van't Veer et al., 2016*). We therefore included unlabelled *meso*-diaminopimelic acid (*m*DAP)-type lipid II into reactions of PBP1B[Ec] with lipid II-Atto550 and lipid II-Atto647n (*Figure 1A*). Both probes were incorporated into the produced PG or glycan chains as indicated by SDS-PAGE analysis (*Figure 1B, I*). After the reaction, fluorescence spectra taken at the excitation wavelength of the donor fluorophore (Atto550, $\lambda_{abs}$=552 nm) showed a reduced donor emission intensity ($\lambda_{fl}$=580 nm) and an increased emission of the acceptor fluorophore (Atto647n, $\lambda_{fl}$=665 nm) (*Figure 1C, I*) indicative of FRET between the two fluorophores.

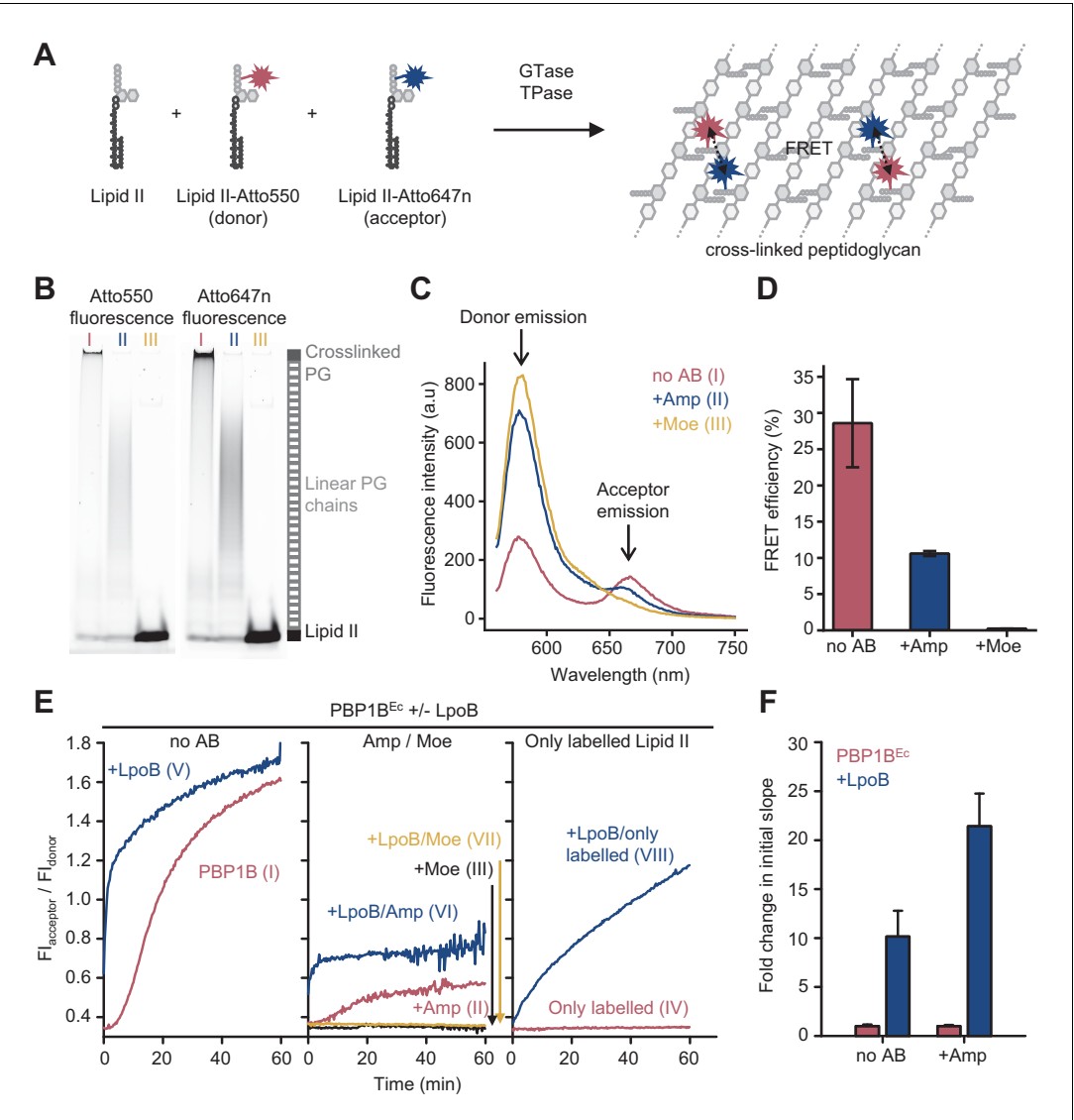

**Figure 1.** Förster resonance energy transfer (FRET) assay to monitor peptidoglycan synthesis in real time. (**A**) Scheme of the reactions of a class A penicillin-binding protein (PBP) (GTase-TPase) with unlabelled lipid II and the two versions of labelled lipid II, yielding a peptidoglycan (PG) product that shows FRET. (**B**) SDS-PAGE analysis of PG products by PBP1B$^{Ec}$ (0.5 μM) reactions with unlabelled lipid II, Atto550-labelled lipid II, and Atto647n-labelled lipid II at a 1:1:1 molar ratio (each 5 μM), in the absence of antibiotics (I, red) or in the presence of 1 mM ampicillin (II, blue) or 50 μM moenomycin (III, yellow). Samples were incubated for 1 hr at 37°C and boiled for 5 min. (**C**) Representative fluorescence emission spectra taken after reactions performed as described in **B** and following the same labelling pattern. (**D**) FRET efficiency for PBP1B$^{Ec}$ reactions carried out as indicated in **B**, calculated using the (ratio)$_A$ method (see Materials and methods). Values are mean ± SD of at least three independent samples. (**E**) Representative reaction curves from FRET assays of detergent-solubilized PBP1B$^{Ec}$. The same components as indicated in **B** were incubated in the presence or absence of 2 μM LpoB(sol). Reactions were performed in the absence of antibiotic (left), with 1 mM ampicillin (Amp) or 50 μM moenomycin (Moe) (middle), or by omitting unlabelled lipid II (right). The numbers indicate the corresponding lane of the gel in *Figure 1—figure supplement 2D*. Samples were incubated for 1 hr at 25°C. (**F**) Averaged initial slopes from reaction curves obtained by the FRET assay for detergent-solubilized *E. coli* PBP1B in the presence (blue) or absence (red) of LpoB, and in the presence or absence of ampicillin. Values are normalized relative to the slope in the absence of activator for each condition and are mean ± SD of 2–3 independent experiments.

The online version of this article includes the following source data and figure supplement(s) for figure 1:

**Source data 1.** Numerical data to support graphs in *Figure 1* and original gel images for *Figure 1B*.
**Figure supplement 1.** Fluorescent lipid II analogues to monitor peptidoglycan synthesis in real time.
*Figure 1 continued on next page*

*Figure 1 continued*

**Figure supplement 2.** Analysis of fluorescence spectra to calculate Förster resonance energy transfer (FRET) efficiency.

**Figure supplement 2—source data 1.** Numerical data to support graphs in *Figure 1—figure supplement 2*.

**Figure supplement 3.** Förster resonance energy transfer assay to monitor peptidoglycan synthesis in real time.

**Figure supplement 3—source data 1.** Original gel images for *Figure 1—figure supplement 3*.

**Figure supplement 4.** Fluorescence intensity (FI) of lipid II-Atto550 and lipid II-Atto647n only changes significantly during reactions when both versions are present.

**Figure supplement 4—source data 1.** Numerical data to support graphs in *Figure 1—figure supplement 4*.

Analysis of the fluorescence spectra allowed to calculate FRET efficiencies which we found to be 29 ± 6% (*Figure 1D*, *Figure 1—figure supplement 2*). Ampicillin, which inhibits the TPase, blocked the formation of crosslinked PG (*Figure 1B, II*) and reduced the FRET efficiency to one third (*Figure 1C, II, D*). Moenomycin, which blocks the GTase, and, indirectly, TPase activities completely abolished the incorporation of fluorescent lipid II and the associated signal (*Figure 1B, III, D*; *Bertsche et al., 2005*). These results demonstrate that incorporation of the labelled probes into PG by PBP1B$^{Ec}$ results in fluorescence energy transfer that depends on the GTase and TPase activity, with the latter being the major contributor.

Next, we monitored reactions in real time by measuring fluorescence emission of the donor and acceptor fluorophores (FI$_{donor}$ and FI$_{acceptor}$, respectively) after excitation of the donor (540 nm) in a microplate reader for 60 min (*Figure 1E*, *Figure 1—figure supplement 3A*). As controls, we also performed reactions containing unlabelled lipid II plus only one of the labelled lipid II versions (lipid II-Atto550 or lipid II-Atto647n) in parallel (*Figure 1—figure supplement 4*). Changes in FI$_{donor}$ and FI$_{acceptor}$ were much higher when both fluorescent lipid II versions were present, in agreement with energy transfer. Thus, we used the ratio between both signals (FI$_{acceptor}$/FI$_{donor}$) as a real-time read-out for FRET and PG synthesis. Without LpoB, FRET appeared after ~5 min and slowly increased until it plateaued after 50–60 min (*Figure 1E*, left). By contrast, reactions with LpoB(sol) showed an immediate and rapid increase in FRET which reached the plateau after 10–20 min, consistent with faster PG synthesis (*Figure 1E*, left). In agreement with the end-point analysis described above, we found no FRET in samples containing moenomycin (*Figure 1E*, middle), and ampicillin generally reduced the final fluorescence ratio level by approximately threefold (*Figure 1E*, middle). Analysis of reaction products by SDS-PAGE also confirmed that crosslinked PG was only produced in the absence of antibiotics, while the presence of ampicillin still allowed the formation of glycan chains (*Figure 1—figure supplement 3B*).

The GTase reaction began after a lag phase, consistent with previously published data (*Schwartz et al., 2002*; *Egan et al., 2014*), which is likely caused by a slower initiation of glycan chain synthesis compared to the rate of polymerization. We measured the slope of FRET reaction curves during the linear raise in signal after the lag phase (when present) and compared the slopes with or without activator. Slopes with LpoB were approximately ten- or twentyfold higher than without the activator, in the absence or presence of ampicillin, respectively (*Figure 1F*). Although this result is comparable to the approximately tenfold activation of the GTase rate by LpoB measured with dansyl-lipid II (*Egan et al., 2014*; *Egan et al., 2018*), a quantification of the individual GTase and TPase reaction rates would require a more exact knowledge of how these two activities contribute to the final FRET signal, which is currently not available (see Discussion).

## Intra-chain versus inter-chain FRET

Because ampicillin substantially reduced the FRET signal, we hypothesized that FRET arises mainly between fluorophores on different glycan chains of a crosslinked PG product (*Figure 1A*). To determine the relative contribution of intra-chain versus inter-chain FRET, we digested PG produced in the presence of labelled lipid II with either the DD-endopeptidase MepM, which cleaves crosslinks between glycan chains (*Singh et al., 2015*; *Singh et al., 2012*), or the muramidase cellosyl, which cleaves the β-(1,4)-glycosidic bond between MurNAc and GlcNAc-producing muropeptides (structures 1–3 in *Figure 2C*; *Rau et al., 2001*; *Figure 2A, B*). As a control, glycan chains produced by PBP1B$^{Ec}$ in the presence of ampicillin were also digested with both hydrolases. SDS-PAGE analysis

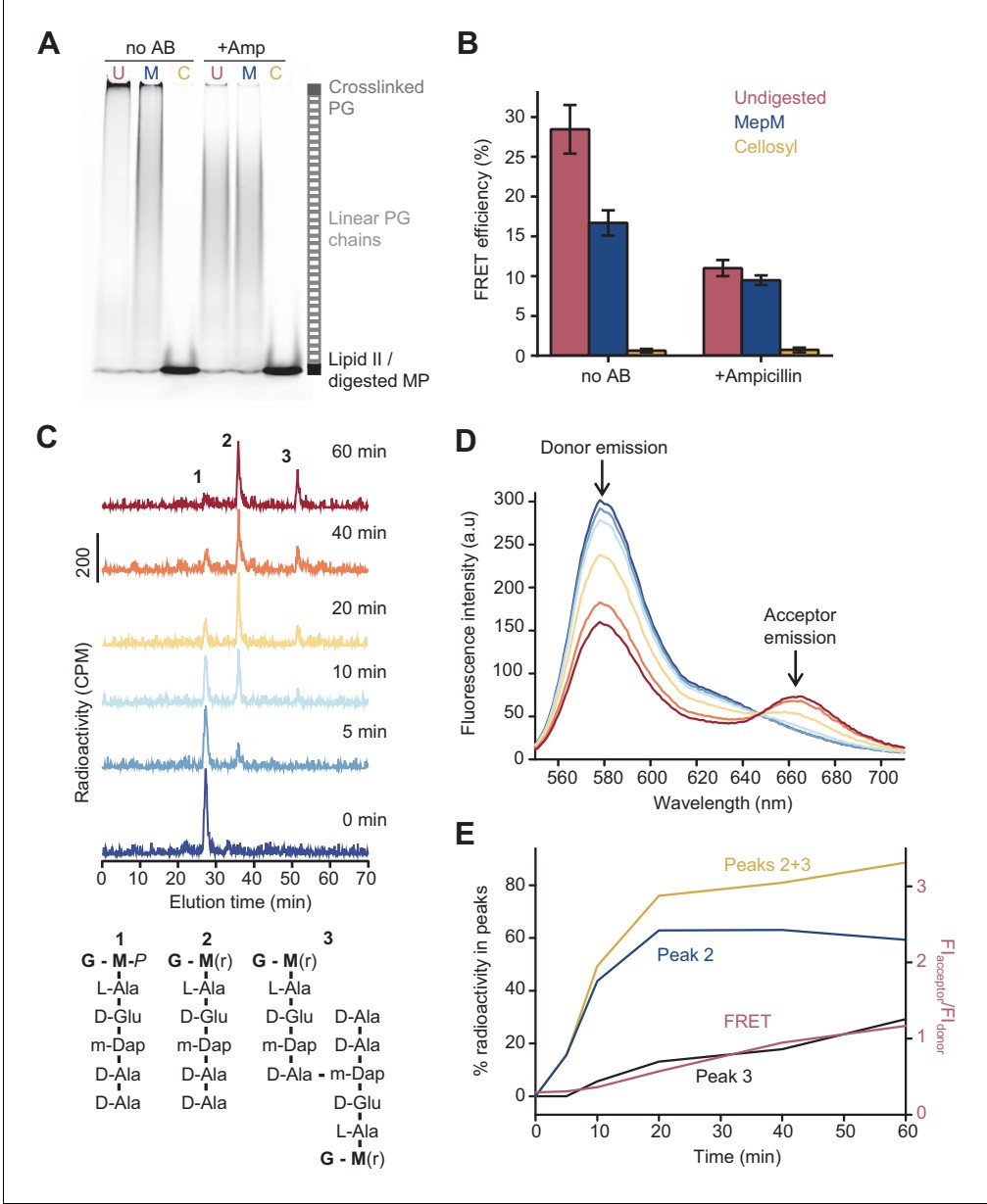

**Figure 2.** The Förster resonance energy transfer (FRET) signal arises from both the glycosyltransferase and transpeptidase reactions. (**A**) Peptidoglycan (PG) synthesized in reactions of PBP1B$^{Ec}$ in the presence or absence of 1 mM ampicillin was incubated with no PG hydrolase (U), DD-endopeptidase MepM (M), or muramidase cellosyl (C), and aliquots were analysed by SDS-PAGE. Reaction conditions were the same as indicated in **Figure 1B–D**. (**B**) FRET efficiency for samples prepared as indicated in **A**, calculated using the (ratio)$_A$ method (see Materials and methods). Values are mean ± SD of at least three independent experiments. (**C**) PBP1B$^{Ec}$ (0.5 µM) was incubated with 5 µM each of lipid II-Atto647n, lipid II-Atto550, and $^{14}$C-labelled lipid II. At indicated time points, aliquots were taken and reactions were stopped by addition of moenomycin. After measuring fluorescence (see **D**), the PG was digested with the muramidase cellosyl, and the resulting muropeptides were reduced with sodium borohydride and separated by HPLC. The structures of muropeptides corresponding to peaks 1–3 are shown below the chromatograms. (**D**) Fluorescence spectra taken with excitation at 522 nm for the samples described in **C**. (**E**) Quantification of peak 2 (GTase product, blue), peak 3 (GTase+TPase, black), or the sum of both 2 and 3 (yellow) from chromatograms in **C**, along with the FRET signal (red) calculated as the ratio of acceptor emission over donor emission from data in **D**.

The online version of this article includes the following source data and figure supplement(s) for figure 2:

**Source data 1.** Numerical data to support graphs in **Figure 2** and original gel images for **Figure 2B**.

*Figure 2 continued on next page*

*Figure 2 continued*

**Figure supplement 1.** Effect of proportion of labelled lipid II substrates on PBP1B$^{Ec}$ activity in detergents.
**Figure supplement 1—source data 1.** Numerical data to support graphs in *Figure 2—figure supplement 1* and original gel images used to quantify labelled lipid II consumption.

confirmed that MepM substantially reduced the amount of crosslinked PG in the samples while cellosyl digested the PG into muropeptides (*Figure 2A*). Next, we measured the FRET efficiency after digestion. MepM digestion had a negligible effect on the FRET efficiency of glycan chains produced in the presence of ampicillin but reduced the FRET efficiency by approximately twofold for crosslinked-PG samples (*Figure 2B*). This confirms that inter-chain FRET is a major contributor to the final FRET signal. MepM did not reduce FRET efficiency to the same value as ampicillin, presumably because of incomplete digestion of the labelled PG. Finally, cellosyl completely abolished FRET for both glycan chains and crosslinked PG (*Figure 2B*).

To confirm that the formation of peptide crosslinks is required to produce substantial FRET in the absence of LpoB, we analysed the PG synthesized by PBP1B$^{Ec}$ from radioactively labelled *m*DAP-type lipid II and the two fluorescent lipid II analogues (*Figure 2C–E*). We monitored the reaction at different time points by fluorescence spectroscopy (FRET measurements) and digested aliquots with cellosyl before separating the resulting muropeptides by HPLC. The monomers and crosslinked muropeptide dimers were quantified by scintillation counting using an in-line radiation detector attached to the HPLC column (*Figure 2C*). FRET increased over time and correlated well with the formation of crosslinked muropeptide dimers, but not the rate of lipid II consumption (peak 2) (*Figure 2D, E*). Overall, we conclude that, in the absence of LpoB, FRET can arise from GTase activity alone (intra-chain FRET), but the overall contribution from the TPase activity (inter-chain FRET) is dominant.

To study in more detail the contribution of intra-chain FRET, we varied the molar fraction of fluorescent lipid II and measured the activity of PBP1B$^{Ec}$ in the presence or absence of activator. Confirming a previous study (*Van't Veer et al., 2016*), PBP1B$^{Ec}$ alone was unable to use lipid II-Atto550 and lipid II-Atto647n for polymerization when unlabelled lipid II was not present (*Figure 1E*, right). Surprisingly, addition of LpoB allowed PBP1B$^{Ec}$ to produce short, non-crosslinked individual PG chains (*Figure 1E*, *Figure 1—figure supplement 3B*) that gave rise to a slow but large increase in FRET (*Figure 1E*, right, *Figure 1—figure supplement 3A*), indicating that polymerization of labelled lipid II occurred in the absence of unlabelled lipid II. To investigate this effect further, we varied the proportion of fluorescent lipid II over non-fluorescent (but radioactive) lipid II and measured the reaction slopes and, at the end of the reaction time, the final FI$_{acceptor}$/FI$_{donor}$ ratio and amounts of unused lipid II versions, in the presence or absence of activator (*Figure 2—figure supplement 1*). LpoB slightly increased consumption of all versions of lipid II by PBP1B$^{Ec}$ (*Figure 2—figure supplement 1A*) but did not affect the proportion of fluorescent material that was incorporated into PG, which reflected the initial percentage of fluorescent lipid II (*Figure 2—figure supplement 1B*). In the absence of LpoB, the final FI$_{acceptor}$/FI$_{donor}$ ratio increased with increasing proportions of labelled lipid II, but this increase was steeper in the presence of LpoB (*Figure 2—figure supplement 1C*). As similar proportions of fluorescent material were incorporated into PG with or without activator, the difference in the final FRET must arise by fluorophores located closer together within the PG produced when LpoB is present. Finally, the reaction slopes did not change significantly with increasing proportions of labelled lipid II in the absence of LpoB but increased in its presence up to 50% of fluorescent lipid II, and then plateaued (*Figure 2—figure supplement 1A*). Overall, these results suggest that LpoB stimulates the incorporation of fluorophores in consecutive positions along the glycan chain and thus increases the contribution of intra-chain FRET. Thus, the increase in slopes observed with activator (*Figure 1F*) reflects not only a higher PG synthesis rate but also a higher contribution of intra-chain FRET.

## FRET assay to monitor PG synthesis in liposomes

To establish the FRET assay for membrane-embedded PG synthases, we reconstituted a version of PBP1B$^{Ec}$ with a single cysteine at the cytoplasmic N-terminus into liposomes prepared from *E. coli* polar lipids (EcPL) (*Figure 3—figure supplement 1A*). The liposome-reconstituted PBP1B$^{Ec}$ became accessible to a sulfhydryl-reactive fluorescent probe only after disrupting the liposomes with

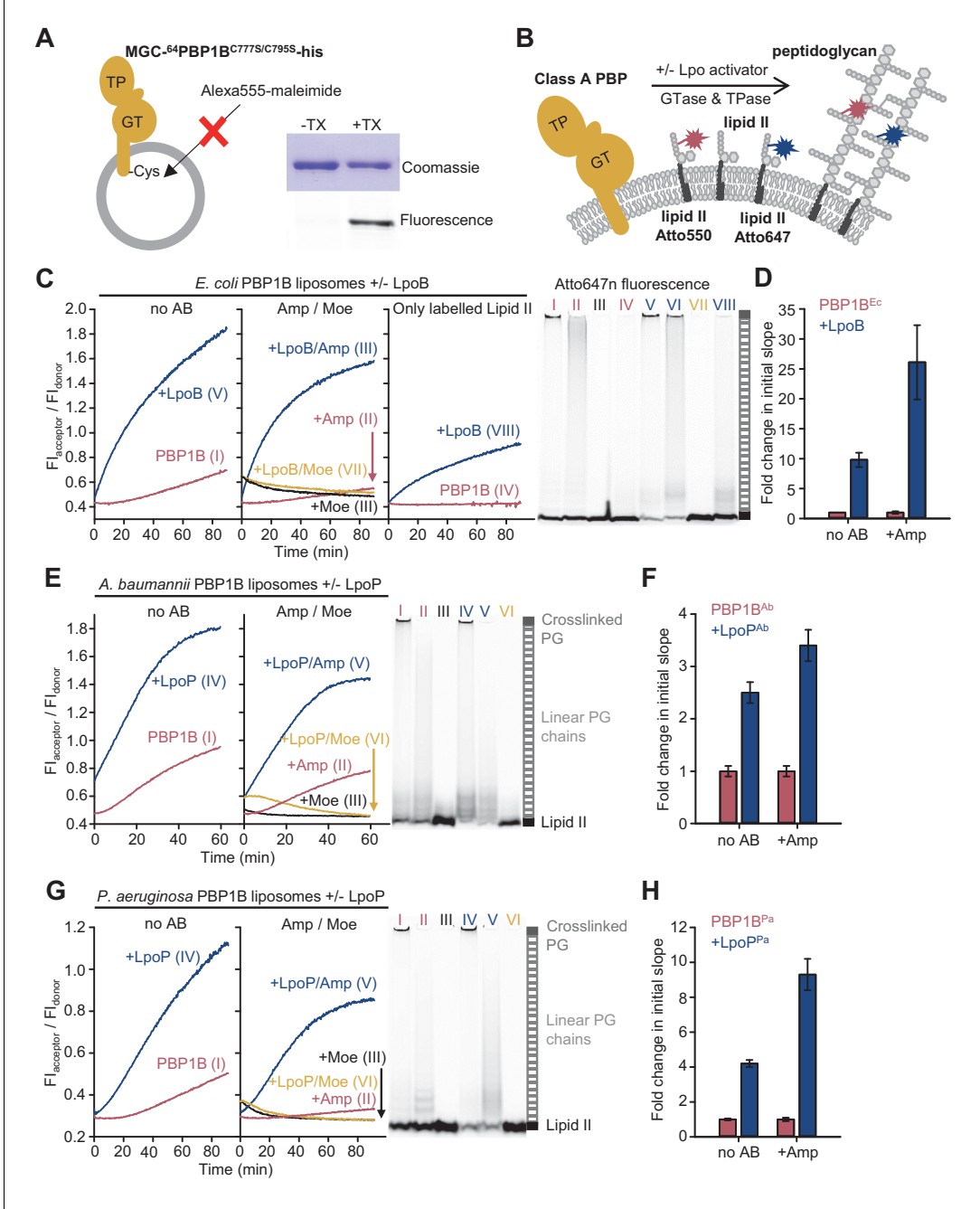

**Figure 3.** The Förster resonance energy transfer (FRET) assay for peptidoglycan synthesis can be adapted for reactions on liposomes. (**A**) Class A penicillin-binding proteins (PBPs) were reconstituted in *E. coli* polar lipid (EcPL) liposomes. To assess the orientation of the liposome-reconstituted PBPs, MGC-$^{64}$PBP1B-his C777S C795S containing a single cysteine in the N-terminal region was reconstituted as in **A**. The accessibility of the cysteine was determined by staining with sulfhydryl-reactive fluorescent probe, Alexa Fluor555-maleimide, in the presence or absence of Triton X-100 (TX). Samples were analysed by SDS-PAGE with fluorescence scanning to detect labelled protein followed by Coomassie staining. (**B**) To perform activity assays in liposomes, class A PBPs were reconstituted along a 1:1 molar ratio mixture of Atto550-labelled lipid II and Atto647n-labelled lipid II in liposomes as in **A**. Reactions were started by addition of unlabelled lipid II in the presence or absence of lipoprotein activators (lpo). Using this methodology, we monitored the activity of PBP1B$^{Ec}$ (**C, D**), PBP1B$^{Ab}$ (**E, F**), and PBP1B$^{Pa}$ (**G, H**). Representative reaction curves are shown. Reactions were carried out in the presence (blue lines) or absence (red lines) of the lipoprotein activators (LpoB(sol) for PBP1B$^{Ec}$, LpoP$^{Ab}$(sol) for PBP1B$^{Ab}$, and LpoP$^{Pa}$(sol) for PBP1B$^{Pa}$), and either in the absence of antibiotic (left) or presence of 1 mM ampicillin (Amp) or 50 μM moenomycin (Moe, black and yellow lines) (middle). For PBP1B$^{Ec}$, control reactions in the absence of unlabelled lipid II (right) are also shown. Products were analysed by SDS-PAGE followed by fluorescence scanning at the end of reactions (right side). Curves are numbered according to the corresponding lane on the SDS-PAGE gels. PBP1B$^{Ec}$, PBP1B$^{Ab}$, and PBP1B$^{Pa}$ were reconstituted in EcPL liposomes containing labelled lipid II (0.5 mol% of lipids, 1:1 molar ratio mixture of

*Figure 3 continued on next page*

*Figure 3 continued*

Atto550-labelled lipid II and Atto647n-labelled lipid II), at protein-to-lipid molar ratios of 1:3000, 1:2000, and 1:3000, respectively. Reactions were started by adding unlabelled lipid II (final concentration 12 µM) and incubated at 37°C for 60 min (PBP1B$^{Ec}$ and PBP1B$^{Ab}$) or 90 min (PBP1B$^{Pa}$) while monitoring fluorescence at 590 and 680 nm with excitation at 522 nm. (D), (F), and (H) show averaged initial slopes from reaction curves obtained by the FRET assay for liposome-reconstituted PBP1B$^{Ec}$, PBP1B$^{Ab}$, and PBP1B$^{Pa}$, respectively, in the presence (blue) or absence (red) of lipoprotein activators and in the presence or absence of ampicillin. Values are normalized relative to the slope in the absence of activator and are mean ± variation of two independent experiments.

The online version of this article includes the following source data and figure supplement(s) for figure 3:

**Source data 1.** Numerical data to support graphs in *Figure 3* and original gel images for *Figure 3C, E and G*.

**Figure supplement 1.** Activity of membrane-reconstituted PBP1B$^{Ec}$ is optimal in *E. coli* polar lipids at low ionic strength.

**Figure supplement 2.** The Förster resonance energy transfer (FRET) assay for peptidoglycan synthesis can be adapted for reactions on liposomes.

**Figure supplement 2—source data 1.** Numerical data to support graphs in *Figure 3—figure supplement 2* and original gel images for *Figure 3—figure supplement 2A*.

**Figure supplement 3.** Moenomycin does not affect Förster resonance energy transfer (FRET) on liposomes with lipid II-Atto550 and lipid II-Atto647n in the absence of class A penicillin-binding proteins.

**Figure supplement 3—source data 1.** Numerical data to support graphs in *Figure 3—figure supplement 3*.

**Figure supplement 4.** Fluorescence intensity (FI) of lipid II-Atto550 and lipid II-Atto647n in the membrane only changes significantly during reactions when both species are present.

**Figure supplement 4—source data 1.** Numerical data to support graphs in *Figure 3—figure supplement 4*.

**Figure supplement 5.** Amino acid sequence comparison between LpoP homologues from *A.baumannii* and *P.aeruginosa*.

**Figure supplement 6.** LpoP$^{Ab}$ stimulates the glycosyltransferase activity of PBP1B$^{Ab}$.

**Figure supplement 6—source data 1.** Numerical data to support graphs in *Figure 3—figure supplement 6*.

**Figure supplement 7.** Peptidoglycan synthesis activity of *A.baumannii* PBP1B in the presence of Triton X-100 followed by Förster resonance energy transfer (FRET).

**Figure supplement 7—source data 1.** Numerical data to support graphs in *Figure 3—figure supplement 7* and original gel images for gels in *Figure 3—figure supplement 7C*.

**Figure supplement 8.** Peptidoglycan synthesis activity of *P. aeruginosa* PBP1B in the presence of Triton X-100 followed by Förster resonance energy transfer (FRET).

**Figure supplement 8—source data 1.** Numerical data to support graphs in *Figure 3—figure supplement 8* and original gel images for gels in *Figure 3—figure supplement 8C*.

detergent (*Figure 3A*), showing that virtually all PBP1B molecules were oriented with the N-terminus inside the liposomes. This suggests that the large, extracellular portion of PBP1B$^{Ec}$ is not transferred through the membrane during the reconstitution into liposomes (*Rigaud and Lévy, 2003*). Next, we reconstituted unmodified PBP1B$^{Ec}$ and tested its activity by adding radioactive lipid II. In contrast to the detergent-solubilized enzyme, the liposome-reconstituted PBP1B$^{Ec}$ required the absence of NaCl from the reaction buffer for improved activity (*Figure 3—figure supplement 1B–E*), suggesting that ionic strength affects either the structure of PBP1B$^{Ec}$ in the membrane, the properties of EcPL liposomes, or the delivery of lipid II into the liposomes.

We next aimed to adapt the FRET assay to study PG synthesis on liposomes to mimic the situation in the cell (*Figure 3*, *Figure 3—figure supplement 2*). As PBP1B$^{Ec}$ did not accept Atto550- or Atto647-derivatized lipid II for GTase reactions in the absence of unlabelled lipid II (*Figure 1E*), we reconstituted PBP1B$^{Ec}$ in liposomes along both Atto-labelled substrates and initiated the reaction by adding unlabelled lipid II (*Figure 3B*). PBP1B$^{Ec}$ reaction rates in liposomes were slower than in the presence of Triton X-100 for all conditions tested (compare curves in *Figure 3C*, measured at 37°C, with the ones in *Figure 1E*, measured at 25°C), and there was a longer lag time before FRET started to increase (*Figure 3C*, left). Moenomycin blocked the increase in FRET, while ampicillin reduced the final FRET levels (*Figure 3C*, middle). For unknown reasons, the FRET signal with moenomycin was initially higher than without moenomycin and then decreased to initial values without moenomycin (*Figure 3C*, middle), independent of the class A PBP used (see below) but not in empty liposomes (*Figure 3—figure supplement 3*). LpoB(sol) produced an approximately tenfold increase in the initial slope, measured as explained above (*Figure 3D*), and the resulting final FRET was much higher (*Figure 3C*, left). In some experiments with PBP1B$^{Ec}$ liposomes in the presence of ampicillin and LpoB(sol), we noticed a slow decrease in FRET after a fast initial increase, and the production of short glycan chains instead of the long chains produced normally (*Figure 3—figure supplement 2A*). As in detergents, without unlabelled lipid II membrane-bound PBP1B produced a FRET signal

only in the presence of LpoB(sol) (*Figure 3C*, right). The analysis of the final products by SDS-PAGE confirmed that both Atto550 and Atto647n were incorporated into glycan chains or crosslinked PG during the reaction in liposomes (*Figure 3C*, right, *Figure 3—figure supplement 2B*). As expected, controls with PBP1B$^{Ec}$ liposomes reconstituted with only lipid II-Atto550 or only lipid II-Atto647n showed significantly lower changes in FI$_{donor}$ and FI$_{acceptor}$ than when both fluorescent versions were present together (*Figure 3—figure supplement 4*).

In summary, using our FRET-based assay we demonstrated real-time monitoring PG synthesis in membrane by PBP1B$^{Ec}$ and showed that the FRET signal was sensitive to the presence of PG synthesis inhibitors (moenomycin and ampicillin).

## Activities of other membrane-bound class A PBPs

To demonstrate the usefulness of the FRET assay to study class A PBPs of potential therapeutic interest, we next tested two PBP1B homologues from Gram-negative pathogens, *Acinetobacter baumannii* (PBP1B$^{Ab}$) and *Pseudomonas aeruginosa* (PBP1B$^{Pa}$). We set up reactions in the presence or absence of a soluble version of the lipoprotein activator LpoP$^{Pa}$(sol) for PBP1B$^{Pa}$ (*Greene et al., 2018*). There is currently no reported activator of PBP1B$^{Ab}$, but next to the gene encoding PBP1B$^{Ab}$ we identified a hypothetical gene encoding a lipoprotein containing two tetratricopeptide repeats (Uniprot code D0C5L6) (*Figure 3—figure supplement 5*) which we subsequently found to activate PBP1B$^{Ab}$ (see below, *Figure 3—figure supplement 6*). We named this protein LpoP$^{Ab}$ and purified a version without its lipid anchor, called LpoP$^{Ab}$(sol). We were able to monitor PG synthesis activity by FRET for both PBPs in the presence or absence of their (hypothetical) activators using the Triton X-100-solubilized (*Figure 3—figure supplements 7* and *8*) or liposome-reconstituted proteins (*Figure 3E–H*, *Figure 3—figure supplement 2D–E*). Our experiments revealed the differences in the activities and the effect of activators between both PBP1B-homolgoues which we discuss in the following paragraphs.

PBP1B$^{Ab}$ showed GTase activity in the presence of Triton X-100 (*Figure 3—figure supplement 6A*) and was stimulated ~3.3-fold by LpoP$^{Ab}$(sol) (*Figure 3—figure supplement 6B*); LpoP$^{Ab}$(sol) also accelerated the consumption of lipid II-Atto550 and glycan chain polymerization (*Figure 3—figure supplement 6C*). We measured a low FRET signal for PG produced by the detergent-solubilized enzyme in the FRET assay (*Figure 3—figure supplement 7A*) and poor production of crosslinked PG (*Figure 3—figure supplement 7C*), unlike in the case of the other PBPs. However, the liposome-reconstituted PBP1B$^{Ab}$ displayed a higher TPase activity than the detergent-solubilized enzyme (compare gels in *Figure 3E*, right, and *Figure 3—figure supplement 7C*). In addition, the final FRET signal was substantially higher in liposomes than in detergents (*Figure 3E*, *Figure 3—figure supplement 7A*). Moenomycin completely blocked FRET development, while ampicillin had a negligible effect on the final FRET levels in detergents and only a small effect in liposomes (~1.2-fold reduction), indicating that intra-chain FRET is the major contributor to FRET (*Figure 3E*, *Figure 3—figure supplement 7A*). LpoP$^{Ab}$(sol) stimulated PBP1B$^{Ab}$, with a higher effect in detergents (12.3-fold increase) than liposomes (~2.5-fold increase) (*Figure 3E, F*, *Figure 3—figure supplement 7A, B*).

PBP1B$^{Pa}$ displayed robust TPase activity in detergents and liposomes (*Figure 3G*, right, *Figure 3—figure supplement 8C*), and ampicillin reduced the final FRET signal by ~1.8-fold in Triton X-100 and ~1.5-fold in liposomes, indicating a substantial contribution of inter-chain FRET to the FRET signal (*Figure 3G*, *Figure 3—figure supplement 8A*). The addition of LpoP$^{Pa}$(sol) resulted in an increase in the final FRET by ~2.2-fold in the membrane and ~2.1-fold in detergents (*Figure 3G*, *Figure 3—figure supplement 8A*), and accelerated initial slopes by ~4.2-fold in the membrane and ~11.5-fold in detergents (*Figure 3H*, *Figure 3—figure supplement 8B*); lipid II consumption was increased under both conditions (*Figure 3G*, right, *Figure 3—figure supplement 8C*). Overall, these results indicate that LpoP$^{Pa}$(sol) stimulates both GTase and TPase activities in agreement with a recent report (*Caveney et al., 2020*).

## PG synthesis on supported lipid bilayers

As we were able to successfully reconstitute active class A PBPs in membranes and monitor their activity in real time, we next aimed to characterize the behaviour of these enzymes in the membrane in more detail by reconstituting them on supported lipid bilayers (SLBs). SLBs are phospholipid bilayers formed on top of a solid support, usually a glass surface, and they allow for studying the

spatial organization of transmembrane proteins and their diffusion along the membrane by fluorescence microscopy at high spatiotemporal resolution.

We optimized the reconstitution of PBP1B[Ec] in SLBs formed with EcPL and used the optimized buffer conditions for activity assays on liposomes. To support lateral diffusion and also improve stability of the proteins incorporated into SLBs, we employed glass surfaces coated with polyethylene glycol (PEG) end-functionalized with a short fatty acid (*Roder et al., 2011*) to anchor the EcPL bilayer (*Figure 4A*). We noticed a decrease in membrane diffusivity and homogeneity at a high surface density of PBP1B[Ec] (*Figure 4—figure supplement 1*). To maintain the integrity of the SLB, we reduced the density of PBP1B[Ec] on SLBs from ~$10^{-3}$ mol protein/mol lipid in liposomes to a range of $10^{-6}$ to $10^{-5}$ mol protein/mol lipid. Using a fluorescently labelled version of PBP1B[Ec] reconstituted in SLBs, we were able to track the diffusion of single PBP1B molecules in the plane of lipid membrane in the

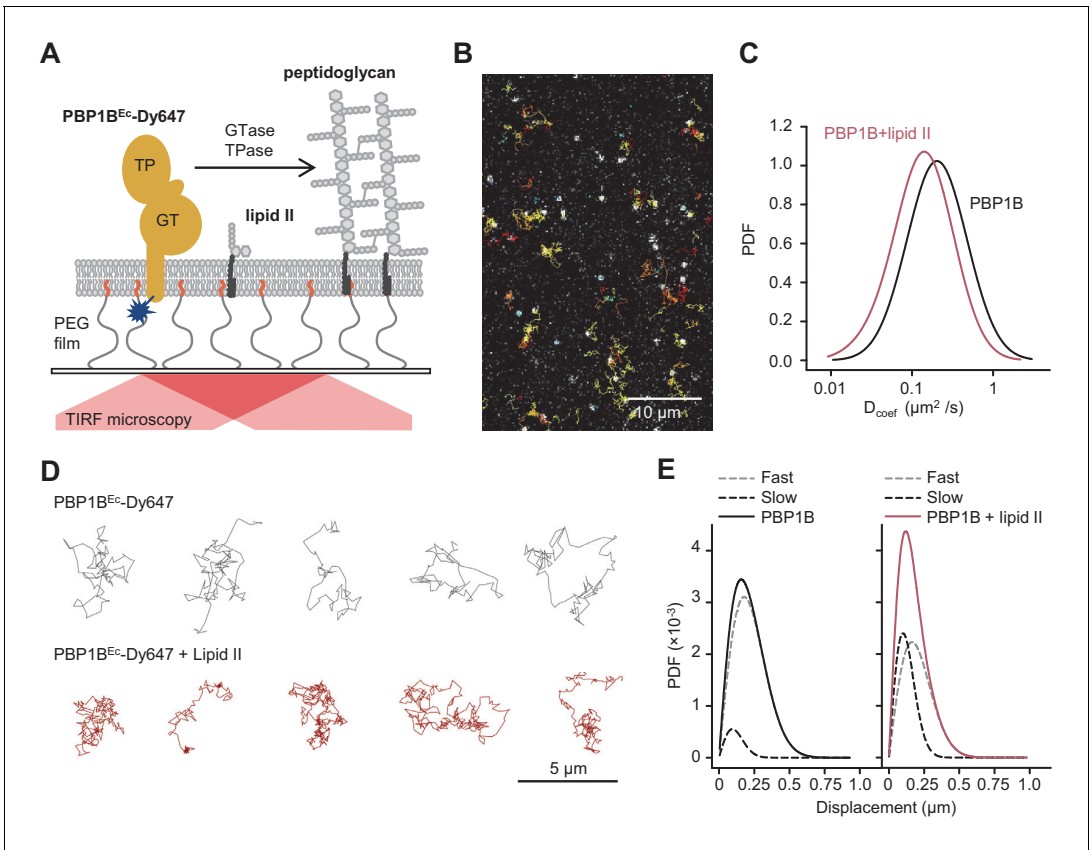

**Figure 4.** Addition of lipid II slows down diffusion of PBP1B on supported lipid bilayers. (**A**) Schematic illustration of the approach (not to scale). A single-cysteine version of PBP1B[Ec] (MGC-[64]PBP1B-his C777S C795S) labelled with fluorescent probe Dy647 in its single Cys residue (PBP1B[Ec]-Dy647) was reconstituted into a polymer-supported lipid membrane formed with *E. coli* polar lipids, and its diffusion was monitored using TIRF microscopy in the presence or absence of substrate lipid II. (**B**) Single-molecule TIRF micrograph of PBP1B[Ec]-Dy647 diffusing in the lipid membrane in the presence of 1.5 µM lipid II (corresponding to *Video 1*). Calculated particle tracks are overlaid. (**C**) Histograms of diffusion coefficients ($D_{coef}$) of PBP1B[Ec]-Dy647 particles in the presence (red) or absence (black) of lipid II. The average $D_{coef}$ decreased from 0.23 ± 0.06 µm²/s to 0.1 ± 0.04 µm²/s upon addition of lipid II. Values are mean ± SD of tracks from three independent experiments. (**D**) Representative tracks for diffusing PBP1B[Ec]-Dy647 particles in the absence (black, top) or presence of lipid II (red, bottom), showing the absence of confined motion in the presence of lipid II. (**E**) Displacement distributions of PBP1B[Ec]-Dy647 particles (solid lines) in the absence (left) or presence (right) of lipid II were analysed using a Rayleigh model incorporating two populations of particles, a fast-diffusing one (grey dashed lines) and a slow-diffusing one (black dashed lines). In the absence of lipid II, only 8 ± 5% of the steps were classified into the slow fraction (121 ± 6 nm average displacement), while the majority of steps were of 257 ± 6 nm (fast fraction). The slow fraction increased upon addition of lipid II to 37 ± 5% of the steps, with an average displacement of 132 ± 16 nm.

The online version of this article includes the following source data and figure supplement(s) for figure 4:

**Source data 1.** Numerical data to support graphs in *Figure 4*.

**Figure supplement 1.** Control of membrane fluidity and integrity upon reconstitution of *E. coli* PBP1B.

**Figure supplement 2.** *E. coli* PBP1B is active after reconstitution in supported lipid bilayers (SLBs).

presence or absence of substrate lipid II by total internal reflection fluorescence (TIRF) microscopy (**Figure 4B, D**, **Video 1**). PBP1B$^{Ec}$ diffused on these supported bilayers with an average $D_{coef}$ of $0.23 \pm 0.06$ μm$^2$/s. Addition of lipid II slowed down PBP1B$^{Ec}$ diffusion (**Figure 4C**), resulting in a lower average $D_{coef}$ of $0.10 \pm 0.06$ μm$^2$/s. Upon addition of lipid II, we could not detect a prolonged confined motion within particle tracks (**Figure 4D**); however, the average length of displacements between two sequential frames was reduced (**Figure 4E**). Thus, we successfully reconstituted diffusing PBP1B$^{Ec}$ in SLBs and observed that lipid II binding slowed down the diffusion of the synthase.

Next, we wanted to confirm that PBP1B$^{Ec}$ remained active to produce planar bilayer-attached PG. We incubated SLBs containing PBP1B$^{Ec}$ with radioactive lipid II and digested any possible PG produced with a muramidase and analysed the digested material by HPLC. Due to the low density and amount of PBP1B$^{Ec}$ on each SLB chamber, we expected a small amount of PG product; hence, we included LpoB(sol) to boost the activity of PBP1B$^{Ec}$. Under these conditions, about 12% of the added radiolabelled lipid II was incorporated into PG after an overnight incubation (**Figure 4—figure supplement 2A**). However, products of both the GTase and TPase activities of PBP1B$^{Ec}$ were detected, and these products were absent in the presence of moenomycin (**Figure 4—figure supplement 2B**). After overnight PG synthesis reactions with radioactive lipid II, about 32% of the radioactivity remained in the membrane fraction after washing (PG products and unused lipid II) and 68% was in the supernatant. The analysis of the membrane and wash fractions by HPLC (**Figure 4—figure supplement 2C, D**) revealed that SLB-reconstituted PBP1B$^{Ec}$ produced crosslinked PG while, importantly, the wash fraction contained no PG products, confirming that the PG synthesis occurred on the SLBs and this PG remained attached to the bilayer. The fraction of membrane-attached radioactivity was almost the same (33%) when PBP1B$^{Ec}$ was not present in the bilayer, indicating that PBP1B$^{Ec}$ did not affect lipid II binding to the bilayer.

## FRET assay on supported bilayers

Next, we adapted the FRET assay to SLBs and TIRF microscopy, taking advantage of the photostability and brightness of the Atto550 and Atto647n probes. Our aim was to visualize PG synthesis by class A PBPs at high resolution as a first step towards understanding PG synthesis at a single molecule level. We used a similar approach as for liposomes, where both Atto550- and Atto647n-labelled lipid II were co-reconstituted with PBP1B$^{Ec}$ on SLBs and PG synthesis was triggered by the addition of unlabelled lipid II (**Figure 3A**). To measure any change in FRET due to PG synthesis, we took advantage of the fact that upon photobleaching of the acceptor probe in a FRET pair the emitted fluorescence intensity of the donor increases as absorbed energy cannot be quenched by a nearby acceptor (**Loose et al., 2011**; **Verveer et al., 2006**). Indeed, we detected an increase in lipid II-Atto550 fluorescence intensity upon photobleaching of the Atto647n probe after the addition of unlabelled lipid II and LpoB(sol), indicating the presence of FRET (**Figure 5A**, **Figure 5—figure supplement 1A**). When we bleached the acceptor at different time points of the reaction, we found the FRET signal to increase after a lag phase of ~8 min. Importantly, there was no FRET increase in the presence of ampicillin (**Figure 5B**, **Figure 5—figure supplement 1A**, **Video 2**) or when a GTase-defective PBP1B$^{Ec}$ version (E233Q) was used (**Figure 5C**). In addition, the FRET signal was abolished when the muramidase cellosyl was added after the PG synthesis reaction (**Figure 5C**). These results imply that the FRET signal detected by microscopy is primarily due to the transpeptidase activity of PBP1B$^{Ec}$, in agreement with the results obtained on liposomes (**Figure 5C**).

## PG synthesized on SLBs

As our experiments confirmed that the PG synthesized by PBP1B$^{Ec}$ on SLBs remained attached to the bilayer, we next analysed the lateral

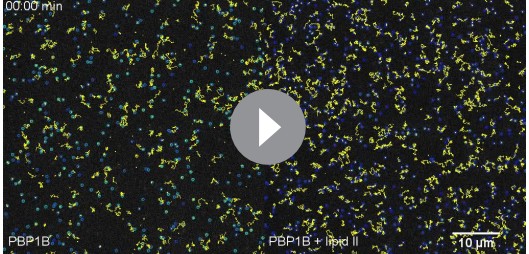

**Video 1.** Single-molecule imaging of PBP1B on supported lipid bilayers (SLBs). PBP1B$^{Ec}$-Dy647 was reconstituted in *E. coli* polar lipids SLBs at a 1:10$^6$ (mol: mol) protein-to-lipid ratio and was tracked using single-molecule TIRF before or after the addition of 1.5 μM lipid II. Images were taken with a rate of 62 ms per frame.

https://elifesciences.org/articles/61525#video1

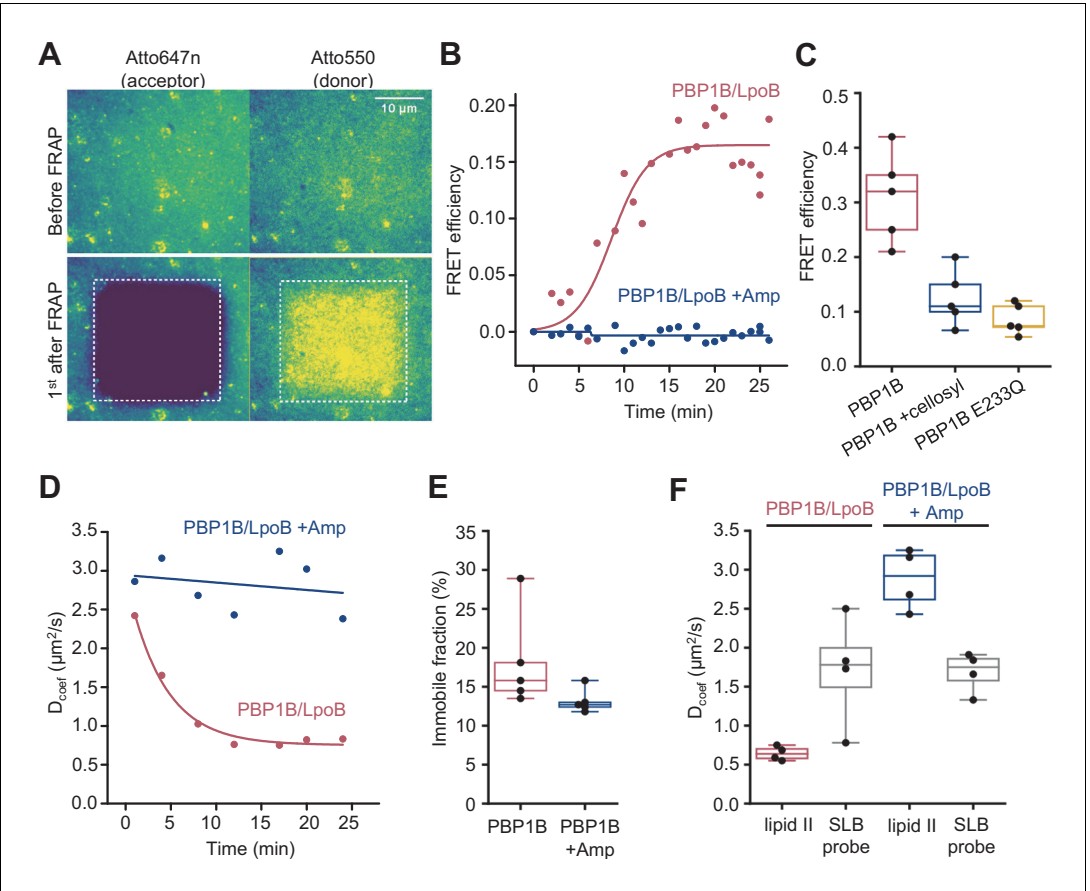

**Figure 5.** Förster resonance energy transfer (FRET) assay on a planar lipid membrane. (A) FRET acquisition by TIRF microscopy. PBP1B[Ec] was reconstituted into a polymer-supported lipid membrane to preserve its lateral diffusion. A supported lipid membrane was formed from *E. coli* polar lipid extract supplemented with 0.5 mol% of labelled lipid II (Atto550 and Atto647n at 1:1 ratio). To initiate peptidoglycan (PG) polymerization, unlabelled lipid II (10 μM) and LpoB(sol) (4 μM) were added from the bulk solution. An increase in FRET efficiency was recorded by dual-colour TIRF microscopy: the acceptor (lipid II-Atto647n) was photobleached, and the concomitant increase in the donor intensity (lipid II-Atto550) was recorded within a delay of 1 s. (B) FRET kinetics of PG polymerization and crosslinking. Inhibition of PBP1B[Ec] TPase activity with 1 mM ampicillin did not produce any changes in the donor intensity, confirming that FRET signal is specific to crosslinked PG. A sigmoid (straight lines) was fitted to the data to visualize the lag in the increase of FRET signal. (C) FRET efficiency was measured after a round of PG synthesis before and after digestion with the muramidase cellosyl. After cellosyl digestion, FRET efficiency decreased by 2.5-fold, resulting in a FRET signal comparable to the one of a control surface with a GTase-defective PBP1B[Ec](E233Q), performed in parallel. Each dot corresponds to a different surface area within the same sample. (D) Quantification of the diffusion coefficient of lipid II-Atto647n over the time course of PG polymerization (left) from the experiment presented in B, calculated from the dynamics of the recovery of lipid II-Atto647n signal within the photobleached region of interest (ROI). (E) Quantification of the fraction of immobile lipid II-Atto647n from several experiments as the one depicted in B; each dot represents the value from a different experiment. (F) Diffusion of lipid II-Atto647n or a phospholipid bound probe labelled with Alexa 488 (supported lipid bilayer) was recorded in a FRAP assay using a 1 s delay and dual-colour imaging, 30 min after initiation of PG synthesis by addition of lipid II and LpoB(sol). Only the diffusion of lipid II, but not of a fluorescently labelled, His$_6$-tagged peptide attached to dioctadecylamine-tris-Ni$^{2+}$-NTA, was affected by the presence of ampicillin during the PG synthesis reaction.

The online version of this article includes the following source data and figure supplement(s) for figure 5:

**Source data 1.** Numerical data to support graphs in *Figure 5*.
**Figure supplement 1.** Control of membrane fluidity and integrity during the Förster resonance energy transfer assay.

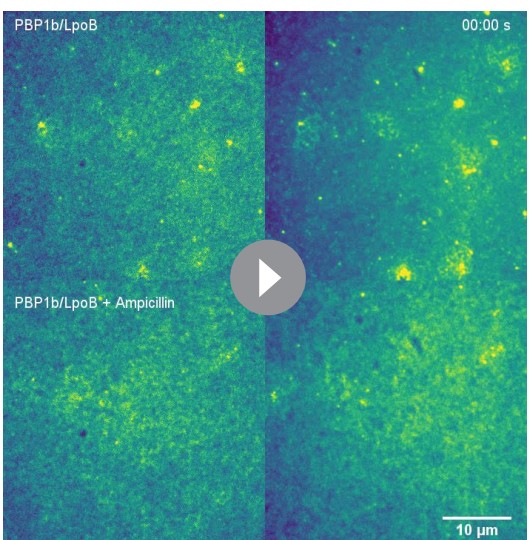

**Video 2.** Förster resonance energy transfer (FRET) assay on supported lipid bilayers (SLBs). PBP1B[Ec] was reconstituted in *E. coli* polar lipids SLBs at a 1:10[5] (mol: mol) protein-to-lipid ratio along lipid II-Atto647 and lipid II-Atto550. Membranes were incubated with 5 µM lipid II in the presence or absence of 1 mM ampicillin. To detect FRET, the fluorescence of the acceptor Atto647n was bleached within a region. In the subsequent frame, the fluorescence of Atto550 increased, indicating the presence of FRET. In the presence of ampicillin, this increase did not happen. https://elifesciences.org/articles/61525#video2

diffusion of lipid II-Atto647n and its products during PG synthesis reactions. We first analysed the recovery of fluorescence intensity after photobleaching to monitor the diffusion of lipid II-Atto647n during PG synthesis (*Figure 5D*). Only when crosslinking was permitted (absence of ampicillin), the diffusion coefficient of lipid II-Atto647n decreased two- to threefold in a time-dependent manner. The time needed to reach the minimum diffusivity value (~10 min) was similar to the lag detected in the increase of FRET efficiency (*Figure 5B*). The fraction of immobile lipid II-Atto647n did not change significantly in the presence or absence of ampicillin ($13 \pm 2\%$ or $18 \pm 6\%$, respectively, p-value=0.15) (*Figure 5E*), indicating that the crosslinked PG was still mobile under these conditions, but diffused more slowly. We also compared the diffusion of lipid II-Atto647n during the PG synthesis reaction with that of an Alexa Fluor 488-labelled membrane-anchored peptide in the presence or absence of ampicillin (*Figure 5F*, *Figure 5—figure supplement 1B*). The inhibition of TPase by ampicillin only affected the diffusivity of lipid II ($2.9 \pm 0.4$ µm$^2$/s with ampicillin and $0.67 \pm 0.1$ µm$^2$/s without), while that of the lipid probe remained unchanged ($1.6 \pm 0.65$ µm$^2$/s with ampicillin and $1.94 \pm 0.62$ µm$^2$/s without). This shows that the membrane fluidity was not altered by the PG synthesis reaction and therefore was not the cause of the change in lipid II diffusivity upon transpeptidation. As the immobile fraction of labelled lipid

II did not increase after PG synthesis and the diffusion was reduced only two- to threefold, we concluded that lipid II-Atto647n was incorporated into small groups of crosslinked glycan chains which can still diffuse on the bilayer.

In summary, we report the incorporation of active PBP1B[Ec] into SLBs, where we could track a decrease in the diffusion of the protein and its substrate during PG synthesis reactions. Using this system, we detected an increase in FRET upon initiation of PG synthesis, only occurring when transpeptidation was not inhibited.

## Discussion

Although class A PBPs are membrane proteins and PG precursor lipid II is embedded in the bilayer, few studies have provided information about the activity of these important enzymes in a membrane environment. Here, we developed a new assay that reports on PG synthesis by these enzymes in detergents, on liposomes, or on SLBs.

### Intra-chain vs. inter-chain FRET

For all PBPs and conditions tested, FRET increased when only the GTase domain was active (i.e., when FRET occurred between probes incorporated along the same strand), but the FRET signal was always higher when transpeptidase was active (*Figures 1–3*, *Figure 3—figure supplements 7* and *8*). For detergent-solubilized PBP1B[Ec], the FRET curve closely followed the rate of the production of crosslinked PG as determined by HPLC analysis of the products (*Figure 2C–E*), and the FRET of PBP1B[Ec]-produced labelled PG decreased substantially upon digestion with an endopeptidase (*Figure 2A, B*). These results indicate that inter-chain FRET (arising from both fluorophores present on different, adjacent glycan chains) was a main component of the total FRET signal. Why is this the

case? FRET depends on the distance and orientation of the two probes. It might be sterically unfavourable that two large Atto550 and Atto647n containing lipid II molecules simultaneously occupy the donor and acceptor sites in the GTase domain (*Van't Veer et al., 2016*), preventing the incorporation of probes (and high FRET) at successive subunits on a single glycan chain. Indeed, for all PBPs tested either in detergents or liposomes, the incorporation of labelled lipid II into glycan chains was more efficient when unlabelled lipid II was present, and, for most enzymes, an activator was required to polymerize glycan chains using labelled lipid II in the absence of unlabelled lipid II. We thus hypothesize that the TPase activity brings glycan chains to close proximity, reducing the distance between probes sufficiently to produce high levels of FRET (*Figure 6*).

## Limitation of the FRET assay

The FRET assay is sensitive and currently the only method that allows to follow PG synthesis continuously in the membrane. Naturally, there are also limitations with the assay. First, the overall FRET signal is a combination from intra-chain and inter-chain FRET, which both depend on the average distances and orientation of the fluorophore molecules on the growing glycan chains. We currently

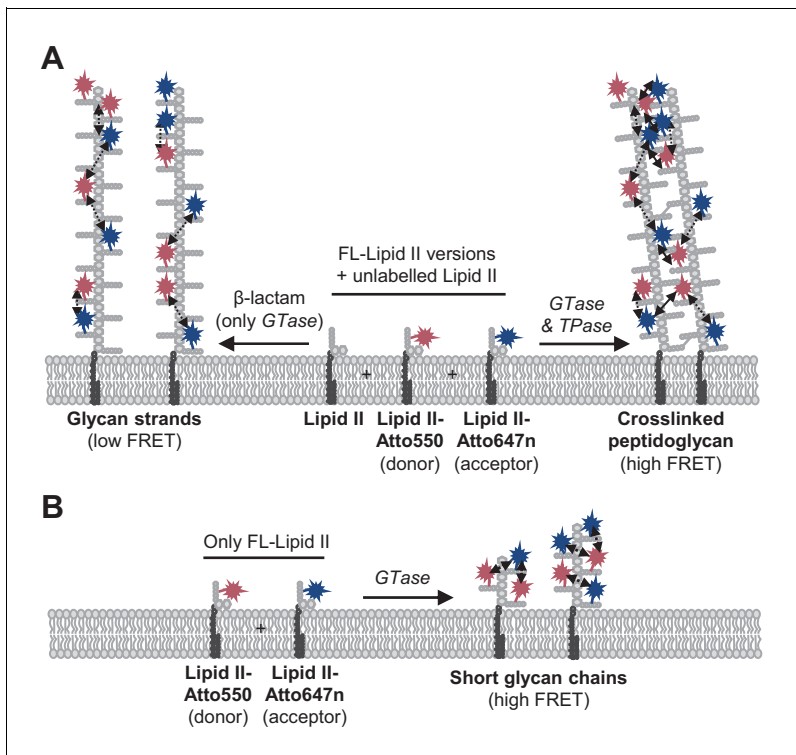

**Figure 6.** Peptidoglycan (PG) synthesis with labelled lipid II versions and detection of Förster resonance energy transfer (FRET). (**A**) A mixture of Atto550-lipid II, Atto647n-lipid II, and unlabelled lipid II is utilized by a class A penicillin-binding protein (PBP) with or without inhibition of the TPase activity by a β-lactam. FRET can only occur between fluorophores within the same glycan strand in linear glycan chains produced in the presence of a β-lactam (left reaction, dashed arrows). When the TPase is active (right reaction), FRET can occur either between probes within the same strand (dashed arrows) or between probes on different strands of the crosslinked PG product (solid arrows). We hypothesize that at any time only one labelled lipid II molecule occupies the two binding sites in the GTase domain and that therefore two probes within the same strand are separated by at least one subunit. As a result, average distances between probes in different strands may be shorter than between probes within the same strand, and thus inter-chain FRET contributes stronger to the total FRET signal than intra-chain FRET. (**B**) Lipoprotein-stimulated PBPs produced short chains when labelled lipid II versions were incubated in the absence of unlabelled lipid II (e.g., *Figure 1B, Figure 1—figure supplement 1C*). In this situation, crosslinking does not occur due to the attachment of the probe to the mDAP residue in the pentapeptide. Within these short strands, intra-chain FRET is stronger than within the long glycan strands depicted in (**A**) due to a shorter average distance between the probes.

do not have a method to measure these parameters individually and determine whether and how they change during the process of PG synthesis, preventing the determination of absolute rates for GTase and TPase reactions. Second, different class A PBPs may produce slightly different distribution and density of fluorophores in the PG synthesized, hence differences in FRET signals may not always reflect different reaction rates. Third, an activator can potentially enhance the ability of a class A PBP to incorporate the fluorescent lipid II analogues, as we observed for LpoB[Ec] and PBP1B[Ec], leading to an increase in intra-chain FRET. Due to these limitations, the assay is inherently semi-quantitative, but with appropriate control samples (β-lactams; only labelled lipid II) it is possible to determine whether the FRET signal follows more the GTase (intra-chain FRET) or TPase (inter-chain FRET) reaction.

## Coupled reactions in class A PBPs and their activation

Our assay revealed the effect of Lpo activators on PBP1B analogues from three bacteria. *P. aeruginosa* uses LpoP to stimulate its PBP1B (*Greene et al., 2018*; *Caveney et al., 2020*). Here, we identified an LpoP homologue in *A. baumannii* and showed that it stimulated its cognate PBP1B. All three PBP1B homologues started the reaction after a lag phase, which was abolished by the addition of the cognate activator (*Egan et al., 2014*; *Caveney et al., 2020*) Considering the recently described role of PBP1B in repairing cell wall defects (*Vigouroux et al., 2020*; *Morè et al., 2019*), the slow start in polymerization and its acceleration by Lpo activators could be an important mechanism to start PG synthesis at gaps in the PG layer where the activators can contact the synthase.

To distinguish the effects of an activator on the TPase and GTase rates requires to use different assays to measure GTase only or GTase/TPase because ongoing glycan chain polymerization is required for transpeptidation to occur (*Bertsche et al., 2005*; *Gray et al., 2015*). An elegant recent report (*Catherwood et al., 2020*) described the use of a coupled D-Ala release assay to determine the kinetic parameters of the TPase activity of PBP1B[Ec] and the effect of LpoB on this rate. Based on their observation that PBP1B[Ec] had barely any TPase activity in the absence of LpoB, the authors concluded that the LpoB-mediated TPase activation explains the essentiality of LpoB for PBP1B function in the cell (*Catherwood et al., 2020*). However, the assay used an enzyme concentration that is too low to support GTase activity in the absence of LpoB, as demonstrated previously (*Pazos et al., 2018*; *Müller et al., 2007*). Therefore, the essentiality of LpoB can be readily explained by its primary effect, the greater than tenfold stimulation of the GTase rate (*Egan et al., 2014*). Our results provide an alternative explanation for PBP1B[Ec] essentiality. Activation by LpoB was much more needed when PBP1B was embedded in the membranes of liposomes and supported bilayers, compared to detergent-solubilized enzyme, supporting the idea that cellular PBP1B strictly requires LpoB for GTase activity. In vitro, LpoB also stimulated the TPase causing PBP1B[Ec] to produce a hyper-crosslinked PG (*Typas et al., 2010*; *Egan et al., 2018*) and the same was observed for LpoP[Pa] and PBP1B[Pa] (*Caveney et al., 2020*). The GTase and TPase contribute both to the signal in our FRET assay, and the relative contribution of intra-chain FRET (due to the GTase) and inter-chain FRET (due to the TPase) can be modified by an activator that enables the incorporation of two adjacent probe molecules on the same glycan chain. Therefore, to untangle the effects of activators on each of the activities requires a single quantitative model accounting for the GTase and TPase rates and including parameters for the initiation, elongation, and termination of glycan chain synthesis of membrane-embedded enzymes. Currently, such a model is not available and our assay could help to develop such a model in the future.

## Class A PBP activities in the membrane

Remarkably, we found slower reaction rates in liposomes than in detergents for all enzymes tested. Several possible factors can explain this, including a slow incorporation of the added unlabelled lipid II into liposomes, a limited capacity of the liposomes to incorporate the unlabelled lipid II, or the accumulation of the undecaprenyl pyrophosphate by-product that has been showed to inhibit PBP1B activity (*Hernández-Rocamora et al., 2018*). None of these factors should change in the presence of LpoB. Hence, we favour the alternative explanation that the membrane-embedding of PBP1B hinders lipid II binding, slowing down the reaction. Remarkably, PBP1B[Ab] showed higher TPase activity in liposomes than in detergents. This observation highlights again that detergents can

affect the activity of membrane proteins and that experimental conditions in PG synthesis assays should be as close as possible at the physiological conditions.

### Towards single-molecule PG synthesis

We also adapted the FRET assay to SLBs and super resolution microscopy to study how PBP1B[Ec] polymerizes PG on SLBs (*Figure 5*). As with the liposome assays, we detected an increase in FRET signal upon triggering PG synthesis that correlated with transpeptidation. Importantly, we could follow the diffusion of the reaction products, which indicates that PBP1B[Ec] does not completely cover the surfaces with a layer of PG but instead produced smaller patches of crosslinked glycan chains. We attribute this to the fact that PBP1B[Ec] was reconstituted at a very low density in order to ensure the homogeneity and stability of the SLBs. Remarkably, we detected a reduction of PBP1B[Ec] diffusivity in the presence of lipid II (*Figure 4*). Previous in vivo single-molecule tracking of fluorescent-protein tagged class A PBPs reported the presence of two populations of molecules, a fast diffusing one and an almost immobile one with a near-zero diffusing rate, which was assumed to be the active population (*Cho et al., 2016*; *Lee et al., 2016*; *Vigouroux et al., 2020*). Our result supports this interpretation, although more experiments are required to further explore this point.

Several real-time methods to study PG synthesis in vitro are described in the literature. However, most of these report on either the GTase or TPase reaction, but not both at the same time, and most available methods are not applicable to the membrane. The scintillation proximity assay by Kumar et al. reports on PG production in a membrane environment and in real time, but it is rather crude in that it uses membrane extract instead of purified protein and relies on the presence of lipid II-synthesizing enzymes present in the extract (*Kumar et al., 2014*). Moreover, it uses radioactivity detection and is not amenable to microscopy, in contrast to methods based on fluorescently labelled substrates. An important advantage of our new assay over other real-time PG synthesis assays is that it uses natural substrates for transpeptidation, that is, nascent glycan strands, instead of mimics of the pentapeptide, and its ability to measure the activities in a natural lipid environment.

Our new FRET assay can potentially be adopted to assay PG synthases in the presence of interacting proteins, for example, monofunctional class B PBPs in the presence of monofunctional GTases (cognate SEDS proteins or Mtg proteins) or interacting class A PBPs (*Meeske et al., 2016*; *Bertsche et al., 2006*; *Sjodt et al., 2020*; *Derouaux et al., 2008*; *Banzhaf et al., 2012*; *Sjodt et al., 2018*; *Taguchi et al., 2019*). In addition, our assay has the potential to be adopted to high-throughput screening for new antimicrobials.

## Materials and methods

**Key resources table**

| Reagent type (species) or resource | Designation | Source or reference | Identifiers | Additional information |
|---|---|---|---|---|
| Strain, strain background (*Escherichia coli*) | BL21(DE3) | New England Biolabs | C2527 | |
| Recombinant DNA reagent | pDML219 | *Bertsche et al., 2006* | | Expression of N-terminal His-tagged *E. coli* PBP1B |
| Recombinant DNA reagent | pKPWV1B | This paper | | Expression of N-terminal His-tagged *Acinetobacter baumannii* 19606 (ATCC) PBP1B |
| Recombinant DNA reagent | pAJFE52 | *Caveney et al., 2020* | | Expression of N-terminal His-tagged *Pseudomonas aeruginosa* PBP1B |
| Recombinant DNA reagent | pMGCPBP1BCS1CS2 | This paper | | Expression of *E. coli* PBP1B version with a single Cys residue in the N-terminus and C-terminal His-tag |

*Continued on next page*

*Continued*

| Reagent type (species) or resource | Designation | Source or reference | Identifiers | Additional information |
|---|---|---|---|---|
| Recombinant DNA reagent | pET28His-LpoB(sol) | *Egan et al., 2014* | | Expression of soluble version of *E. coli* LpoB with an N-terminal His-tag |
| Recombinant DNA reagent | pKPWVLpoP | This paper | | Expression of N-terminal His-tagged *A. baumannii* 19606 (ATCC) LpoP |
| Recombinant DNA reagent | pAJFE57 | *Caveney et al., 2020* | | Expression of soluble version of *P. aeruginosa* LpoP with an N-terminal His-tag |
| Sequence-based reagent | PBP1B.Acineto-NdeI_f | This paper | PCR cloning primers | AGATATCATATGATGAAGTT TGAACGTGGTATC GGTTTCTTC |
| Sequence-based reagent | PBP1B.Acineto-BamHI_r | This paper | PCR cloning primers | GCGGGATCCTTAGTTGTTA TAACTACCACTTGA AATG |
| Sequence-based reagent | Seq1_rev_PBP1B_Acineto | This paper | PCR cloning primers | AGGTTCTAAACGGGCAACTC |
| Sequence-based reagent | Seq2_fwd_PBP1B_Acineto | This paper | PCR cloning primers | TGGTTATGGATTGGCCTCTC |
| Sequence-based reagent | Seq3_fwd_PBP1B_Acineto | This paper | PCR cloning primers | CTGGGCAAGCCAGATTGAAG |
| Sequence-based reagent | Seq4_fwd_PBP1B_Acineto | This paper | PCR cloning primers | ACAATTACGCCAGACACCAG |
| Sequence-based reagent | PBP1B-MGC-F | This paper | PCR cloning primers | CATCATCCATGGGCTGTGGCT GGCTATGGCTACTGCTA |
| Sequence-based reagent | PBP1B-CtermH-R | This paper | PCR cloning primers | CATCATCTCGAGATTAC TACCAAACATATCCTT |
| Sequence-based reagent | C777S-D | This paper | PCR mutagenesis primers | AACTTTGTTTCCAGCGGTGGC |
| Sequence-based reagent | C777S-C | This paper | PCR mutagenesis primers | GCCACCGCTGGAAACAAAGTT |
| Sequence-based reagent | C795S-D | This paper | PCR mutagenesis primers | CAATCGCTGTCCCAGCAGAGC |
| Sequence-based reagent | C795S-C | This paper | PCR mutagenesis primers | GCTCTGCTGGGACAGCGATTG |
| Chemical compound | [14C]Glc*N*Ac-labelled lipid II (*m*DAP) | *Breukink et al., 2003 Bertsche et al., 2005* | | |
| Chemical compound | Lipid II (*m*DAP) | *Egan et al., 2015* | | |
| Chemical compound | Lipid II (Lys) | *Egan et al., 2015* | | |
| Chemical compound | Lipid II-dansyl | *Egan et al., 2015* | | |
| Chemical compound | Lipid II-Atto550 | *Mohammadi et al., 2014 Van't Veer, 2016* | | |
| Chemical compound | Lipid II-Atto647n | *Mohammadi et al., 2014 Van't Veer, 2016* | | |
| Chemical compound | Polar lipid extract from *E. coli* (EcPL) | Avanti Polar Lipids | 100600P | |
| Chemical compound | 1,2-Dioleoyl-*sn*-glycero-3-phosphocholine (DOPC) | Avanti Polar Lipids | 850375P | |
| Chemical compound | 1-Palmitoyl-2-oleoyl-*sn*-glycero-3-phospho-(1'-rac-glycerol) (POPG) | Avanti Polar Lipids | 840457P | |
| Chemical compound | Tetraoleoyl cardiolipin | Avanti Polar Lipids | 710335P | |
| Chemical compound | Dy647P1-maleimide probe | Dyomics | 647P1-03 | |

*Continued on next page*

*Continued*

| Reagent type (species) or resource | Designation | Source or reference | Identifiers | Additional information |
|---|---|---|---|---|
| Chemical compound | Alexa Fluor 488 C5 Maleimide | ThermoFisher Scientific | A10254 | |
| Chemical compound | Alexa Fluor 555 C2 maleimide | ThermoFisher Scientific | A20346 | |
| Chemical compound | Triton X-100 | Roche | 10789704001 | |
| Chemical compound | Moenomycin | Sigma | 32404 | |
| Chemical compound | Ampicillin | Sigma | A9518 | |
| Chemical compound | Methyl-β-cyclodextrin | Sigma-Aldrich | 332615 | |
| Chemical compound | Poly(ethylene glycol) $M_n 8000$ | Sigma-Aldrich | **1546605** | |
| Chemical compound | 1,2-dioleoyl-*sn*-glycero-3-phosphoethanolamine-N-(lissamine rhodamine B sulfonyl) (DOPE-Rhodamine) | Avanti Polar Lipids | **810150C** | |
| Chemical compound | Dioctadecylamine (DODA)-tris-Ni-NTA | *Beutel et al., 2014* | | |
| Chemical compound | cOmplete, EDTA-free Protease Inhibitor Cocktail | Roche Molecular Biochemicals | 5056489001 | |
| Chemical compound | Phenylmethylsulfonylfluoride (PMSF) | Sigma-Aldrich | P7626 | |
| Chemical compound | Ni-NTA superflow resin | Qiagen | 1018142 | |
| Chemical compound | Bio-Beads SM-2 resin | Bio-Rad | 1523920 | |
| Commercial assay, kit | Pierce BCA Protein Assay Kit | ThermoFisher Scientific | 23227 | |
| Commercial assay, kit | HiTrap SP HP column, 1 mL | GE biosciences | 17115101 | |
| Commercial assay, kit | HiTrap Desalting column, 5 mL | GE biosciences | 17140801 | |
| Commercial assay, kit | Prontosil 120–3 C18 AQ reversed-phase column | BISCHOFF Chromatography | 1204F184P3 | |
| Peptide, recombinant protein | DNase | ThermoFisher Scientific | 90083 | |
| Peptide, recombinant protein | Cellosyl | Hoechst (Germany) | | Mutanolysin from Sigma (M9901) can also be used |
| Peptide, recombinant protein | MepM | Federico Corona, following protocol in *Singh et al., 2012* | | |
| Chemical compound | His6-tagged (on the C-terminus) neutral peptide | BioMatik | | CMSQAALNTRNSEEEVSS RRNNGTRHHHHHH |
| Software, algorithm | Fiji | | https://fiji.sc | |
| Software, algorithm | Matlab | MathWorks | https://www.mathworks.com | |
| Software, algorithm | frap_analysis | *Jönsson, 2020* | | |

## Chemicals

[$^{14}$C]Glc*N*Ac-labelled lipid II and the lysine or *m*DAP forms of lipid II were prepared as published (*Breukink et al., 2003*; *Egan et al., 2015*). Lipid II-Atto550 and Lipid II-Atto647n were prepared from the lysine form of lipid II, as described previously (*Egan et al., 2015*), and Atto550-alkyne or Atto647n-alkyne (Atto tec, Germany) in two steps: (1) conversion of lysine form of lipid II to

azidolysine form and (2) labelling of azidolysine lipid II via click-chemistry. The protocol is extensively detailed elsewhere (*Mohammadi et al., 2014*). The advantage of using this methodology over directly attaching the probes to the amine group is the higher yield of click-chemistry reactions, allowing the use of a smaller excess of the reactive florescent probes (*Van't Veer et al., 2016*). All lipid II variants were kept in 2:1 chloroform:methanol at −20℃. Before enzymatic assays, the required amounts of lipid II were dried in a speed-vac and resuspended in water (for assays in detergents) or the appropriate buffer (for liposome and SLB assays). Polar lipid extract from *E. coli* (EcPL), 1,2-dioleoyl-*sn*-glycero-3-phosphocholine (DOPC), 1-palmitoyl-2-oleoyl-*sn*-glycero-3-phospho-(1'-*rac*-glycerol) (POPG), and tetraoleoyl cardiolipin (TOCL) were obtained from Avanti Polar Lipids (USA). Lipids were resuspended in chloroform:methanol (2:1) at a concentration of 20 g/L, aliquoted, and stored at −20℃. Triton X-100, ampicillin, phenylmethylsulfonyl fluoride (PMSF), protease inhibitor cocktail (PIC), and β-mercaptoethanol were from Merck. n-Dodecyl-beta-D-maltopyranoside was purchased from Anatrace (USA). Moenomycin was purchased from Hoechst, Germany. All other chemicals were from Merck.

## Cloning

### Construction of overexpression vector pKPWV1B

The plasmid pKPWV1B was constructed for overexpression of full-length *A. baumannii* PBP1B (PBP1B$^{Ab}$: aa 1–798) with a cleavable N-terminal oligo-histidine tag (His$_6$ tag). Therefore, the gene *mrcB* was amplified using the Phusion high-fidelity DNA polymerase and the oligonucleotides PBP1B.Acineto-NdeI_f and PBP1B.Acineto-BamHI_r and genomic DNA of *A. baumannii* 19606 (ATCC) as template. The resulting PCR fragment and the plasmid DNA of the overexpression vector pET28a(+) (Novagen) were digested with *Nde*I and *Bam*HI, ligated, and transformed into chemical-competent *E. coli* DH5α cells with kanamycin selection. Plasmid DNA of transformants was isolated and sent for sequencing using the following oligonucleotides: Seq1_rev_PBP1B_Acineto, Seq2_fwd_PBP1B_Acineto, Seq3_fwd_PBP1B_Acineto, and Seq4_fwd_PBP1B_Acineto.

### Construction of overexpression vector pKPWVLpoP

The sequence of the hypothetical PBP1B activator of *A. baumannii* 19606 (LpoP$^{Ab}$: NCBI reference number: WP_000913437.1) contains a TPR fold and was found by blast analysis through its homology to *P. aeruginosa* LpoP (30% identity). The plasmid pKPWVLpoP was purchased from GenScript. The gene was synthesized without the first 51 nucleotides (encoding the 17 amino acids of the signal peptide) and with codon optimization for overexpression in *E. coli*. The codon-optimized gene was subcloned in the overexpression vector pET28a(+) using the cloning sites *Nde*I and *Bam*HI, enabling the overexpression of the protein with an N-terminal oligo-histidine tag.

### MGC-$^{64}$PBP1B-his C777S/C795S

This fusion protein contains PBP1B with the substitution of the N-terminal cytoplasmic tail for residues MGC and the addition of a hexahistine tag at the C-terminus. To obtain this construct, the regions coding for amino acids 64 to 844 of PBP1B were amplified from genomic DNA using oligonucleotides PBP1B-MGC-F and PBP1B-CtermH-R. The resulting product was cloned into pET28a+ vector (EMD Biosciences) after digestion with NcoI and XhoI. C777S and C795S mutations were introduced using the QuikChange Lightning mutagenesis kit (Agilent) through oligonucleotide primers C777S-D, C777S-C, C795S-D, and C795S-C. The resulting plasmid was called pMGCPBP1BCS1CS2.

## Purification and labelling of proteins

The following proteins were purified following published protocols: PBP1B$^{Ec}$ (*Bertsche et al., 2006*), LpoB(sol) (*Egan et al., 2014*), PBP1B$^{Pa}$ (*Caveney et al., 2020*), LpoP$^{Pa}$(sol) (*Caveney et al., 2020*), and MepM (*Singh et al., 2012*). All chromatographic steps were performed using an AKTA Prime-Plus system (GE Healthcare).

### *E. coli* PBP1B

The protein was expressed as a fusion with an N-terminal hexahistidine tag in *E. coli* BL21(DE3) pDML924 grown in 4 L of autoinduction medium (LB medium supplemented with 0.5% glycerol,

0.05% glucose, and 0.2% α-lactose) containing kanamycin at 30°C for ~16 hr. Cells were harvested by centrifugation (10,000 × $g$, 15 min, 4°C) and the pellet resuspended in 80 mL of buffer I (25 mM Tris-HCl, 1 M NaCl, 1 mM EGTA, 10% glycerol, pH 7.5) supplemented with 1× PIC (Sigma-Aldrich), 100 µM PMSF (Sigma-Aldrich), and DNase I. After disruption by sonication on ice, membrane fraction was pelleted by centrifugation (130,000 × $g$ for 1 hr at 4°C) and resuspended in buffer II (25 mM Tris-HCl, 1 M NaCl, 10% glycerol, 2% Triton X-100, pH 7.5) by stirring at 4°C for 24 hr. Extracted membranes were separated from insoluble debris by centrifugation (130,000 × $g$ for 1 hr at 4°C) and incubated for 2 hr with 4 mL of Ni$^{2+}$-NTA beads (Novagen) equilibrated in buffer III (25 mM Tris-HCl, 1 M NaCl, 20 mM imidazole, 10% glycerol, pH 7.5). Beads were washed 10 times with 10 mL of buffer III, and the protein was eluted with 3 mL buffer IV (25 mM Tris-HCl, 0.5 M NaCl, 20 mM imidazole, 10% glycerol, pH 7.5). His-PBP1B-containing fractions were pooled and treated with 2 U/mL of thrombin (Novagen) for 20 hr at 4°C during dialysis against dialysis buffer I (25 mM Tris-HCl, 0.5 M NaCl, 10% glycerol, pH 7.5). Protein was then dialyzed in preparation for ion exchange chromatography, first against dialysis buffer II (20 mM sodium acetate, 0.5 M NaCl, 10% glycerol, pH 5.0), then against dialysis buffer II with 300 mM NaCl, and finally against dialysis buffer II with 100 mM NaCl. Finally, the sample was applied to a 1 mL HiTrap SP column (GE Healthcare) equilibrated in buffer A (20 mM sodium acetate, 100 mM NaCl, 10% glycerol, 0.05% reduced Triton X-100, pH 5.0). The protein was eluted with a gradient from 0% to 100% buffer B (as A, with 2 M NaCl) over 14 mL PBP1B-containing fractions that were pooled and dialyzed against storage buffer (20 mM sodium acetate, 500 mM NaCl, 10% glycerol, pH 5.0) and stored at −80°C.

### A. baumannii 19606 PBP1B

The protein was expressed in *E. coli* BL21 (DE3) freshly transformed with plasmid pKPWV1B using the same protocol as PBP1B$^{Ec}$. Cells were harvested by centrifugation (6,200 × $g$ for 15 min at 4°C) and resuspended in 120 mL of PBP1B$^{Ab}$ buffer I (20 mM NaOH/H$_3$PO$_4$, 1 M NaCl, 1 mM EGTA, pH 6.0) supplemented with DNase I, PIC (1:1000 dilution), and 100 µM PMSF. After disruption by sonication on ice, the membrane fraction was pelleted by centrifugation (130,000 × $g$ for 1 hr at 4°C) and resuspended in PBP1B$^{Ab}$ extraction buffer (20 mM NaOH/H$_3$PO$_4$, 1 M NaCl, 10% glycerol, 2% Triton X-100, pH 6.0) supplemented with PIC and PMSF by stirring at 4°C for 16 hr. Extracted membranes were separated from insoluble debris by centrifugation (130,000 × $g$ for 1 hr at 4°C) and incubated with 4 mL of Ni$^{2+}$-NTA beads equilibrated in PBP1B$^{Ab}$ extraction buffer containing 15 mM imidazole. Beads were washed 10 times with 10 mL of PBP1B$^{Ab}$ wash buffer (20 mM NaOH/H$_3$PO$_4$, 10% glycerol, 0.2% Triton X-100, 1 M NaCl, 15 mM imidazole, pH 6.0), and the protein was eluted with 3 mL buffer IV PBP1B$^{Ab}$ elution buffer (20 mM NaOH/H$_3$PO$_4$, 10% glycerol, 0.2% Triton X-100, 1 M NaCl, 400 mM Imidazole, pH 6.0).

PBP1B$^{Ab}$-containing fractions were pooled and dialyzed in preparation for ion exchange chromatography, first against PBP1B$^{Ab}$ dialysis buffer I (20 mM sodium acetate, 1 M NaCl, 10% glycerol, pH 5.0), then against PBP1B$^{Ab}$ dialysis buffer II (20 mM sodium acetate, 300 mM NaCl, 10% glycerol, pH 5.0), and finally against PBP1B$^{Ab}$ dialysis buffer III (10 mM sodium acetate, 100 mM NaCl, 10% glycerol, pH 5.0). The sample was centrifuged for 1 hr at 130,000 × $g$ and 4°C, and the supernatant was applied to a 5 mL HiTrap SP HP column equilibrated in PBP1B$^{Ab}$ buffer A (20 mM sodium acetate, 100 mM NaCl, 10% glycerol, 0.2% Triton X-100, pH 5.0). The protein was eluted from 0% to 100% PBP1B$^{Ab}$ buffer B (20 mM sodium acetate, 2 M NaCl, 10% glycerol, 0.2% Triton X-100, pH 5.0) over 70 mL. PBP1B$^{Ab}$-containing fractions were pooled and dialyzed against PBP1B$^{Ab}$ storage buffer (10 mM sodium acetate, 500 mM NaCl, 0.2% Triton X-100, 20% glycerol, pH 5.0) and stored at −80°C.

### P. aeruginosa PBP1B

The protein was expressed on *E. coli* BL21(DE3) freshly transformed with plasmid pAJFE52, which encodes PBP1B$^{Pa}$ as a fusion with an N-terminal hexahistidine tag in *E. coli* BL21(DE3). Cells were grown in 4 L of LB at 30°C, and expression was induced for 3 hr with 1 mM isopropyl β-D-1-galactopyranoside (IPTG) when the culture reached an OD$_{578}$ of 0.6. PBP1B$^{Pa}$ was extracted and purified using the same protocol as for *E. coli* PBP1B, with the exception that only 2 mL of Ni$^{2+}$ beads were used.

## MGC-[64]PBP1B-his C777S/C795S

This protein was expressed in *E. coli* BL21(DE3) freshly transformed with plasmid pMGCPBP1BCS1CS2 and subsequently purified using the same protocol as for the WT protein, except for the addition of 1 mM tris(2-carboxyethyl)phosphine (TCEP) to all purification buffers. The protein was labelled with Dy647-maleimide probe (Dyomics, Germany) following the manufacturer's instructions. Briefly, 10.2 μM protein was incubated with 100 μM probe and 0.5 mM TCEP for ~20 hr at 4°C, and free probe was removed by desalting using a 5 mL HiTrap desalting column (GE Healthcare).

## LpoB(sol)

The protein was expressed on *E. coli* BL21(DE3) transformed with pET28His-LpoB(sol). Cells were grown in 1.5 L of LB plus kanamycin at 30°C to an $OD_{578}$ of 0.4–0.6, and expression was induced with 1 mM of IPTG for 3 hr at 30°C. Cells were pelleted and resuspended in buffer I (25 mM Tris-HCl, 10 mM $MgCl_2$, 500 mM NaCl, 20 mM imidazole, 10% glycerol, pH 7.5) plus DNase, PIC, and PMSF. Cells were disrupted by sonication on ice and centrifuged (130,000 × *g*, 1 hr, 4°C) to remove debris. The supernatant was applied to a 5 mL HisTrap HP column (GE Healthcare) equilibrated in buffer I. After washing with buffer I, the protein was eluted with a stepwise gradient with buffer II (25 mM Tris-HCl, 10 mM $MgCl_2$, 500 mM NaCl, 400 mM imidazole, 10% glycerol, pH 7.5). Fractions containing the protein were pooled and the His-tag was removed by addition of 2 U/mL of thrombin while dialyzing against buffer IEX-A (20 mM Tris-HCl, 1000 mM NaCl, 10% glycerol, pH 8.3). Digested protein was applied to a 5 mL HiTrap Q HP column (GE Healthcare) at 0.5 mL/min. LpoB (sol) was collected in the flow through, concentrated, and applied to size exclusion on a Superdex200 HiLoad 16/600 column (GE Healthcare) at 1 mL/min in a buffer containing 25 mM HEPES-NaOH, 1 M NaCl, 10% glycerol at pH 7.5. Finally, the protein was dialyzed against storage buffer (25 mM HEPES-NaOH, 200 mM NaCl, 10% glycerol at pH 7.5) and stored at −80°C.

## *A. baumannii* 19606 LpoP(sol)

The protein was expressed on *E. coli* BL21(DE3) transformed with plasmid pKPWVLpoP. Cells were grown overnight at 30°C in 4 L of autoinduction medium. Cells were pelleted by centrifugation (6200 × *g* for 15 min at 4°C) and resuspended in 80 mL of buffer I (25 mM Tris/HCl, 10 mM $MgCl_2$, 1 M NaCl, 20 mM imidazole, pH 7.5) supplemented with DNase I, PIC (1:1000 dilution), and 100 μM PMSF. Cells were disrupted by sonication on ice and centrifuged (130,000 × *g* for 1 hr at and 4°C) to removed debris. The supernatant was incubated for 1 hr with 6 mL Ni-NTA beads preequilibrated in buffer I at 4°C with gentle stirring. The resin was split in two columns, each washed 10 times with 5 mL wash buffer (25 mM Tris/HCl, 10 mM $MgCl_2$, 1 M NaCl, 20 mM imidazole, pH 7.5), and the protein was eluted 7 times with 2 mL of elution buffer (25 mM Tris/HCl, 10 mM $MgCl_2$, 1 M NaCl, 400 mM imidazole, pH 7.5). The best fractions according to SDS-PAGE analysis were pooled and dialyzed stepwise against increasing percentage of dialysis buffer I (25 mM HEPES/NaOH, 10 mM $MgCl_2$, 200 mM NaCl, 10% glycerol, pH 7.5). Thrombin (nine units) was added to the protein to cleave the N-terminal $His_6$ tag overnight at 4°C. The successful cleavage of the N-terminal $His_6$ tag was confirmed by SDS-PAGE. The protein was diluted 2× with 25 mM HEPES/NaOH, 10 mM $MgCl_2$, 10% glycerol, pH 7.5 to reduce the amount of NaCl down to 100 mM. The protein was applied to a 5 mL HiTrap SP HP column and washed with buffer A (25 mM HEPES/NaOH, 10 mM $MgCl_2$, 100 mM NaCl, 10% glycerol, pH 7.5). The protein was then eluted with a gradient of 100 mM to 1 M NaCl over 50 mL at 1 mL/min using increasing percentage of buffer B (25 mM HEPES/NaOH, 10 mM $MgCl_2$, 1 M NaCl, 10% glycerol, pH 7.5). Fractions were collected and analysed by SDS-PAGE. The best fractions were pooled, dialyzed against 25 mM HEPES/NaOH, 200 mM NaCl, 10% glycerol, 10 mM $MgCl_2$, pH 7.5, and the protein was stored at −80°C.

## *P. aeruginosa* LpoP(sol)

The protein was expressed on *E. coli* BL21(DE3) freshly transformed with plasmid pAJFE57, encoding $His_6$-LpoP$^{Pa}$(sol). Cells were grown on 1.5 L LB at 30°C to an $OD_{578}$ of 0.5, and expression was induced for 3 hr by addition of 1 mM IPTG. After harvesting, cells were resuspended in 80 mL of 25 mM Tris-HCl, 500 mM NaCl, 20 mM imidazole, 10% glycerol at pH 7.5. After addition of PIC and 100 μM PMSF, cells were disrupted by sonication on ice. Debris was removed by centrifugation

(130,000 × *g*, 1 hr, 4°C) and the supernatant was applied to a 5 mL HisTrap column equilibrated in resuspension buffer. After washing with 25 mM Tris-HCl, 1 M NaCl, 40 mM imidazole, 10% glycerol at pH 7.5, the protein was eluted with 25 mM Tris-HCl, 500 mM NaCl, 400 mM imidazole, 10% glycerol at pH 7.5. Fractions containing His-LpoP$^{Pa}$(sol) were pooled and the His-tag was removed by addition of 4 U/mL of thrombin while dialyzing against 20 mM Tris-HCl, 200 mM NaCl, 10% glycerol at pH 7.5 for 20 hr at 4°C. The sample was concentrated and further purified by size exclusion column chromatography at 0.8 mL/min using a HiLoad 16/600 Superdex 200 column equilibrated in 20 mM HEPES-NaOH, 200 mM NaCl, 10% glycerol at pH 7.5. LpoP$^{Pa}$-containing fractions that were pooled, concentrated, aliquoted, and stored at −80°C.

## PG synthesis assays in the presence of detergents

### In vitro PG synthesis assay using radiolabelled lipid II in detergents

To assay the in vitro PG synthesis activity of PBP1B$^{Ec}$ with radiolabelled lipid II substrate in the presence of detergents, we used a previously published assay (*Banzhaf et al., 2012*; *Biboy et al., 2013*). Final reactions included 10 mM HEPES/NaOH pH 7.5, 150 mM NaCl, 10 mM MgCl$_2$, and 0.05% Triton X-100. The concentration of PBP1B$^{Ec}$ was 0.5 µM. Reactions were carried out for 1 hr at 37°C. Reactions were stopped by boiling for 5 min. Digestion with cellosyl, reduction with sodium borohydride, and analysis by HPLC were performed as described (*Biboy et al., 2013*).

### FRET-based in vitro PG synthesis assay in detergents

For assays in detergents, samples contained 50 mM HEPES/NaOH pH 7.5, 150 mM NaCl, 10 mM MgCl$_2$, and 0.05% Triton X-100 in a final volume of 50 µL. PBP1B$^{Ec}$, PBP1B$^{Ab}$, or PBP1B$^{Pa}$ were added at a concentration of 0.5 µM. When indicated, activators LpoB(sol), LpoP$^{Ab}$(sol), or LpoP$^{Pa}$(sol) were added at a concentration of 2 µM. Reactions were started by the addition of an equimolar mix of lipid II, lipid II-Atto550, and lipid II-Atto647n, each at 5 µM and monitored by measuring fluorescence using a Clariostar plate reader (BMG Labtech, Germany) with excitation at 540 nm and emission measurements at 590 and 680 nm. In controls containing unlabelled lipid II plus only one of the labelled lipid II versions (lipid II-Atto550 or lipid II-Atto647n) (*Figure 1—figure supplement 4*), the labelled lipid II was added at 5 µM along 10 µM of unlabelled lipid II. Reactions were incubated at the indicated temperature for 60 or 90 min. After the reaction, emission spectra from 550 to 740 nm were taken in the same plate reader with excitation at 522 nm. When indicated, ampicillin was added at 1 mM and moenomycin was added at 50 µM. After plate reader measurements, reactions were stopped by boiling for 5 min, vacuum-dried using a speed-vac desiccator, and analysed by Tris-Tricine SDS-PAGE as described previously (*Van't Veer et al., 2016*).

FRET reactions in the presence of radiolabelled lipid II described in *Figure 1E, F* were performed using the same buffer and substrate and enzyme concentrations as for the plate reader assay but in a final volume of 350 µL. Samples were incubated at 25°C with shaking using an Eppendorf Thermomixer. Also, 50 µL aliquots were taken out at the indicated times and reactions were stopped by addition of 100 µM moenomycin. Samples were then transferred to a 96-well plate to measure FRET as described above. Finally, samples were transferred back to Eppendorf tubes, digested with cellosyl, and reduced with sodium borohydride as described previously (*Biboy et al., 2013*).

### Quantification of lipid II consumption after the FRET assay in detergents

For the assay in *Figure 2—figure supplement 1*, reactions were performed in the same conditions as described for the FRET assay in detergents, but four different molar proportions of fluorescent lipid II over the total amount of lipid II were used: 80%, 66.7%, 50%, and 20%. Radiolabelled lipid II was included to allow for quantification of non-fluorescent lipid II consumption at the end of reactions. The total concentration of lipid II was kept at 15 µM, and the molar ratio of lipid II-Atto550 to lipid II-Atto647n was 1:1. At each proportion of fluorescent lipid II, PBP1B$^{Ec}$ reactions were measured in triplicate with and without LpoB (2 µM), and two control reactions with substrate but no enzyme were prepared to determine the amount of each type of lipid II consumed. Reactions were monitored using a plate reader for 1 hr at 25°C as described above and then stopped by boiling for 10 min. Next, 5 µL aliquots were taken from each reaction, dried in a speed-vac, and analysed by SDS-PAGE with fluorescent scanning as described previously (*Van't Veer et al., 2016*). Fluorescent lipid II consumption was calculated by comparing the intensity of the lipid II bands on reaction lanes

and control lanes. To quantify the consumption of radiolabelled lipid II, the unused lipid II was extracted from the remaining aliquot of the reactions with butanol:pyridine (2:1, vol:vol) at pH 4.2, and the radioactivity was quantified by scintillation counting as described previously (*Egan et al., 2015*).

## Analysis of FRET reaction curves

Reaction curves were obtained by calculating the ratio between fluorescence intensity at 680 and 590 nm monitored at every well. This maximizes the amount of information captured from the change in the spectrum due to FRET and normalizes the intensity removing any non-specific jumps in the signal due to bubbles in the reaction well or lamp instability. The slope of reaction curves obtained by the FRET assay was calculated when the ratio started to rise, avoiding the lag phase when present. Only the linear phase of each curve was used. For example, for PBP1B$^{Ec}$ in detergents, slopes were calculated from 10 to 15 min in the absence of LpoB and within the first minute in the presence of activator. To compare our results with prior reports, we report the fold-change in the slope in the presence of the corresponding Lpo activator, that is, the ratio between the slope in a condition with activator and the slope at the same condition without activator.

## Determination of FRET efficiency

For the determination of FRET efficiency, reactions were prepared in the same conditions as for the plate reader assays but they were incubated at 37°C for 1 hr in Eppendorf tubes instead and boiled for 5 min afterwards. For every antibiotic condition, four samples were prepared: sample 1 (DA reaction) contained 5 µM each of unlabelled lipid II, lipid II-Atto550, and lipid II-Atto647n; sample 2 (D reaction) contained 10 µM unlabelled lipid II plus 5 µM lipid II-Atto550; sample 3 (A reaction) contained 10 µM unlabelled lipid II plus 5 µM lipid II-Atto647n; and sample 4 (BG reaction) contained 15 µM unlabelled lipid II. For digestion with hydrolases, 50 µL of the PG synthesis reactions were prepared as described above and split into three aliquots. Either 5 µM MepM, 0.05 mg/mL cellosyl, or buffer was added to a final volume of 20 µL, and samples were incubated overnight at 37°C and boiled for 5 min to stop reactions.

Samples were measured in Cary Varian fluorimeter using a 1.5 mm light-path quartz cuvette. For samples 1, 3, and 4, two spectra were measured, one with excitation at 552 nm ($\lambda_{ex}^D$) and emission collected from 560 to 750 nm (*ds* spectrum) and the other with excitation at 650 nm ($\lambda_{ex}^A$) and emission collected from 660 to 750 nm (*as* spectrum). For sample 2, only the *ds* spectrum was measured. All spectra were taken with 5 nm slits for emission and excitation at the same detector voltage settings (850 V).

FRET efficiency (*E*) was calculated according to the (ratio)$_A$ method described in *Vámosi and Clegg, 1998*. Briefly, (ratio)$_A$ is a normalized measure of the enhancement of the acceptor emission due to FRET,

$$(\text{ratio})_A = \frac{F_A(\lambda_{ex}^D, \lambda_{em})}{F_A(\lambda_{ex}^A, \lambda_{em})} = \frac{\left[\epsilon_D(\lambda_{ex}^D)E + \epsilon_A(\lambda_{ex}^D)\right]\Phi^A(\lambda_{em})}{\epsilon_A(\lambda_{ex}^A)\Phi^A(\lambda_{em})} = \frac{\epsilon_D(\lambda_{ex}^D)E + \epsilon_A(\lambda_{ex}^D)}{\epsilon_A(\lambda_{ex}^A)} \tag{1}$$

where $\Phi^A(\lambda_{em})$ is a shape function of the acceptor emission spectrum, $F_A(\lambda_{ex}^D, \lambda_{em})$ is the emission of the acceptor (only the acceptor) when excited at $\lambda_{ex}^D$, and $F_A(\lambda_{ex}^A, \lambda_{em})$ is the emission of the acceptor when excited at $\lambda_{ex}^D$, both in the sample containing both donor and acceptor. FRET efficiency is normalized by the extinction coefficients of the donor at $\lambda_{ex}^D$ ($\epsilon_D(\lambda_{ex}^D)$ = 120,000 M$^{-1}$ cm$^{-1}$) and of the acceptor at both $\lambda_{ex}^D$ and $\lambda_{ex}^A$ ($\epsilon_A(\lambda_{ex}^D)$ = 6000 M$^{-1}$ cm$^{-1}$, and $\epsilon_A(\lambda_{ex}^A)$ = 150,000 M$^{-1}$ cm$^{-1}$, respectively).

In order to calculate (ratio)$_A$ from the *ds* spectrum, three spectral contributions ($\delta^{ds}$, $\alpha^{ds}$, and $\beta^{ds}$) were fitted in the *ds* spectra, $F^{ds}(\lambda)$, and two spectral contributions ($\alpha^{as}$ and $\beta^{as}$) were fitted in the *as* spectra, $F^{as}(\lambda)$:

$$F^{ds}(\lambda) = \delta^{ds}F_{Dref}^{ds}(\lambda) + \alpha^{ds}F_{Aref}^{ds}(\lambda) + \beta^{ds}F_{Bref}^{ds}(\lambda) \tag{2}$$

$$F^{as}(\lambda) = \alpha^{as}F_{Aref}^{as}(\lambda) + \beta^{as}F_{Bref}^{as}(\lambda) \tag{3}$$

where $F_{Dref}^{ds}(\lambda)$ is the background-free spectra from the donor-only reference sample exited at $\lambda_{ex}^D$; $F_{Aref}^{ds}(\lambda)$ is the background-free spectra from the acceptor-only reference sample exited at $\lambda_{ex}^D$; $F_{Aref}^{as}(\lambda)$ is the background-free spectra from the acceptor-only reference sample exited at $\lambda_{ex}^A$; and $F_{Bref}^{ds}(\lambda)$ and $F_{Bref}^{as}(\lambda)$ are the background spectra obtained at $\lambda_{ex}^D$ and $\lambda_{ex}^A$, respectively. (ratio)$_A$ was then calculated from Equation 4, integrating at wavelengths common in both $F_{Aref}^{ds}(\lambda)$ and $F_{Aref}^{ds}(\lambda)$ (from 660 to 750 nm).

$$(\text{ratio})_A = \frac{\alpha^{ds} F_{Aref}^{ds}(\lambda)}{\alpha^{as} F_{Aref}^{as}(\lambda)} \tag{4}$$

All calculations were implemented in Excel.

## Continuous GTase assay using dansylated lipid II

Continuous fluorescence GTase assays using dansylated lipid II and *A. baumannii* PBP1B were performed as described previously (*Schwartz et al., 2001*; *Offant et al., 2010*; *Egan and Vollmer, 2016*). Samples contained 50 mM HEPES/NaOH pH 7.5, 105 mM NaCl, 25 mM MgCl$_2$, 0.039% Triton X-100, and 0.14 µg/µL cellosyl muramidase in a final volume of 60 µL. PBP1B$^{Ab}$ was added at a concentration of 0.5 µM. When indicated, LpoP$^{Ab}$(sol) was added at a concentration of 0.5 µM. Reactions were started by addition of dansylated lipid II to a final concentration of 10 µM and monitored by following the decrease in fluorescence over 60 min at 37°C using a FLUOstar OPTIMA plate reader (BMG Labtech, Germany) with excitation at 330 nm and emission at 520 nm. The fold-increase in GTase was calculated against the mean rate obtained with PBP1B$^{Ab}$ alone at these reaction conditions, at the fastest rate.

## Time-course GTase assay by SDS-PAGE followed by fluorescence detection

PBP1B$^{Ab}$ at a concentration of 0.5 µM was incubated with 5 µM lipid II-Atto550 and 25 µM unlabelled lipid II in the presence or absence of 1.5 µM LpoP$^{Ab}$(sol). Reactions contained 20 mM HEPES, 150 mM NaCl, 10 mM MgCl$_2$, 0.06% TX-100, and 1 mM ampicillin to block transpeptidation. Aliquots were taken after 0, 2, 5, 10, 30, and 60 min incubation at 37°C, boiled for 10 min to stop reactions, and analysed by Tris-Tricine SDS-PAGE followed by fluorescence detection as described previously (*Van't Veer et al., 2016*).

## Fluorescence scanning of SDS-PAGE gels with PG synthesis products

SDS-PAGE gels were scanned using either a Typhoon FLA9500 (GE) or a Typhoon 9400 (Amersham) fluorescence scanner. Atto550 fluorescence was scanned using a 532 nm laser and either a 590 nm, 30 nm-bandwidth bandpass filter (9400) or a 575 nm long-pass filter (FLA9500). Atto647n fluorescence was scanned using a 635 nm laser and either a 670 nm, 30 nm-bandwidth bandpass filter (9400) or a 665 nm long-pass filter (FLA9500). Voltage of the photodetector was carefully adjusted to avoid saturated pixels in the resulting images. For quantification, ImageJ was used utilizing the original files produced by the scanner. For visualization, images were exported using ImageQuant software (GE Healthcare). During this conversion, ImageQuant automatically adjusts contrast so that tenuous bands are easier to visualize. Unfortunately, this adjustment makes the most intense bands (unused lipid II, wells) appear saturated, which can be misleading when interpreting the images. Thus, we provide the original gel files produced by the scanner as source data files.

## PG synthesis in liposomes

### Reconstitution of class A PBPs in liposomes

Proteoliposomes containing class A PBPs were prepared as described previously with some modifications (*Egan et al., 2015*; *Rigaud and Lévy, 2003*; *Hernández-Rocamora et al., 2018*). The appropriate lipid or mixture of lipids was dried in a glass test tube under stream of N$_2$ to form a lipid film followed by desiccation under vacuum from 2 hr. When labelled lipid II was co-reconstituted with the indicated class A PBP, they were added at 1:200 mol:mol phospholipid to each lipid II-Atto550 and lipid II-Atto647n. Resuspension into multilamellar vesicles (MLVs) was achieved by addition of 20 mM Tris/HCl, pH 7.5 with or without 150 mM NaCl as indicated in each experiment and several

cycles of vigorous mixing and short incubations in hot tap water. The final lipid concentration was 5 g/L. To form large unilamellar vesicles (LUVs), MLVs were subjected to 10 freeze–thaw cycles and then extruded 10 times through a 0.2 µm filter. LUVs were destabilized by the addition of Triton X-100 to an effective detergent:lipid ratio of 1.40 and mixed with proteins in different protein-to-lipid molar ratios (1:3000 for PBP1B[Ec] and PBP1B[Pa], and 1:2000 for PBP1B[Ab]). After incubation at 4°C for 1 hr, prewashed adsorbent beads (Biobeads SM2, BioRad, USA; 100 mg per 3 µmol of Triton X-100) were added to the sample to remove detergents. Biobeads were exchanged after 2 and 16 hr, followed by incubation with fresh Biobeads for a further 2 hr. After removal of Biobeads by short centrifugation at $4000 \times g$, liposomes were pelleted at $250,000 \times g$ for 30 min at 4°C. The pellet containing proteoliposomes was resuspended using the appropriate buffer. The resuspension was done in a 43% smaller volume than the volume added of lipid II, so that the final concentration of lipids was 11.6 g/L. Samples were then centrifuged for 5 min at $17,000 \times g$ and 4°C to remove any possible aggregates. The supernatant was then used in the appropriate assays. Liposomes were analysed by SDS-PAGE and, only for liposomes without labelled lipid II, also by bicinchoninic acid assay (Pierce BCA Assay Kit, ThermoFisher Scientific, USA) to determine protein concentration. The concentration of protein for liposomes with labelled lipid II was calculated by densitometry of the samples in SDS-PAGE gels after reactions were carried out.

## PBP1B[Ec] orientation assay

To assess the orientation of liposome-reconstituted PBP1B[Ec], MGC-[64]PBP1B-his C777S C795S mutant containing a single cysteine in the N-terminal region was reconstituted in liposomes with EcPL as described above. The accessibility of the cysteine was determined using sulfhydryl-reactive fluorescent probe Alexa Fluor555-maleimide. Reactions containing 0.5 µM protein, 10 µM Alexa Fluor555-maleimide, and 0.2 mM TCEP were incubated for 16 hr at 4°C in the presence or absence 0.5% Triton X-100. Reactions were stopped by addition of 5 mM DTT and boiling for 5 min. Samples were loaded in a 10% acrylamide gel and, after electrophoresis, gels were first scanned using an Amersham Typhoon Trio with excitation at 533 nm and a 40 nm-wide band-pass emission filter at 580 nm. The gel was then stained by Coomassie.

## In vitro PG synthesis assay using radiolabelled lipid II in liposomes

The same methodology as in detergents was used to assay the in vitro PG synthesis activity of PBP1B[Ec] in liposomes, with minor modifications. To start reactions, 1.5 nmol [$^{14}$C]-labelled lipid II were dried in a 0.5 mL glass tube using a vacuum concentrator, resuspended in 5 µL of the appropriate liposome buffer, and mixed with liposomes, buffer, and $MgCl_2$ to a total volume of 50 µL. Final reactions contained 0.5 µM PBP1B[Ec], 30 µM lipid II, and 1 mM $MgCl_2$ in 20 mM Tris/HCl pH 7.5 with or without 150 mM NaCl as indicated for each experiment. Samples were incubated for 90 min at 37°C with shaking at 800 rpm. Reactions were stopped by boiling for 5 min. Digestion with cellosyl, reduction with sodium borohydride, and analysis by HPLC were performed as described (*Biboy et al., 2013*).

## FRET-based in vitro PG synthesis assay in liposomes

For assays with liposomes, samples contained 20 mM Tris pH 7.5, 1 mM $MgCl_2$ in a final volume of 50 µL. In this case, the same volume for each liposome preparation was added to the reactions, 10 µL, so that the total amount of labelled lipid II was present in every reaction. In these conditions, concentration of lipid II-Atto550 and lipid II-Atto647n would be 14.5 µM each, assuming no loss of lipids during sample preparation. The final concentrations of enzymes for the reactions shown in *Figure 3*, determined by densitometry of SDS-PAGE gels, were ~0.59 µM for PBP1B[Ec], ~0.81 µM for PBP1B[Ab], and ~0.53 µM for PBP1B[Pa]. When indicated, activators LpoB(sol), LpoP[Ab](sol), or LpoP[Pa](sol) were added at a concentration of 2 µM. Reactions were started by the addition of lipid II at 12 µM and monitored by measuring fluorescence over a period of 60 min (or 90 min for PBP1B[Pa] liposomes) at 37°C using a Clariostar plate reader (BMG Labtech, Germany), with emission measurements at 590 and 680 nm after excitation at 522 nm. When indicated, ampicillin was added at 1 mM and moenomycin was added at 50 µM. Activity assays were performed immediately after preparation of liposomes was finished as we noticed that some proteins could slowly start polymerization

using the labelled lipid II. After reactions, samples were analysed by Tris-Tricine SDS-PAGE as indicated for detergents.

## Analysis of FRET reaction curves

Slopes were calculated as indicated in the FRET assay in the presence of detergent. As it is not possible to precisely adjust the final amount of enzyme in different liposome preparations, there could be differences in activities measured due to different enzyme amounts. Therefore, we calculated the ratio of the slope with activator over the slope without activator for every liposome preparation and then averaged the values (instead of averaging the different measurements from every sample). At least two independent liposome preparations were prepared for every class A PBP.

## Assays in SLBs

### Preparation of small unilamellar vesicles (SUVs) and proteoliposomes for SLB formation

Liposomes of *EcPL* lipids and proteoliposomes with reconstituted PBP1B$^{Ec}$ were prepared as described previously by addition of β-cyclodextrin to the solution of lipids and Triton X-100 detergent (*Degrip et al., 1998*; *Roder et al., 2011*). Briefly, a thin lipid film of EcPL extract was prepared by N$_2$-assisted chloroform evaporation. After 2 hr of drying under vacuum, the lipid film was rehydrated to 5 mM (total phosphorus concentration) in 150 mM NaCl, 10 mM Tris-HCl, pH 7.4 supplemented with 20 mM Triton X-100. The suspension of lipids/detergent was extensively vortex, freeze/thawed for five cycles, and sonicated using a water-bath sonicator for 10 min (on ice, to avoid lipids overheating upon sonication). To prepare proteoliposomes, full-length PBP1B produced as described above and containing 0.05% Triton X-100 was mixed with a lipid–detergent suspension at the indicated ratio, usually 1:25,000 (protein:lipids), and incubated for 10 min at room temperature (RT). Incorporation of PBP1B$^{Ec}$ into liposomes was achieved by addition of 2× excess of β-cyclodextrin solution for 5 min (at RT) with subsequent twentyfold dilution in 20 mM HEPES, pH 7.4. The rapid depletion of detergent by addition of β-cyclodextrin leads to the formation of very small unilamellar vesicles with an average diameter of 18–25 nm and narrow size distribution (*Roder et al., 2011*).

To prepare liposomes with fluorescently labelled lipid II, the extract of EcPL was supplemented with 2 mol% solution of either lipid II-Atto550 or lipid II-Atto647n. The lipid film was treated similarly as the film for the preparation of proteoliposomes. Liposomes were also prepared by cyclodextrin-assisted extraction of Triton X-100.

### Formation of polymer-SLBs and reconstitution of PBP1B$^{Ec}$ into a supported lipid membrane

To form polymer-supported lipid membranes, the coverslips were functionalized beforehand with a dense PEG film, where the ends of the polymer brush were covalently modified with palmitic acid, which served as a linker to capture liposomes as described elsewhere (*Roder et al., 2011*). To perform a FRET assay on supported lipid membrane, empty EcPL liposomes (1), liposomes with 2 mol% of either lipid II-Atto550 (2) or lipid II-Atto647n (3), and PBP1B$^{Ec}$ proteoliposomes (4) were mixed at equimolar ratio and diluted by twentyfold with the 10 mM Tris pH 7.5 buffer directly in the reaction chamber. After 30 min of incubation at 37°C, the reaction chamber was washed five times by solution exchange. Proteoliposomes adsorbed on the surface were fused by the addition of 10% (w/v) PEG 8 kDa solution (in water). The fusion reaction was carried out for 15 min at 37°C, afterwards PEG solution was rigorously washed out. Fluidity and homogeneity of the lipid membrane were checked either with PE-Rhodamine dye (Avanti) or by addition of a His$_6$-tagged (on the C-terminus) neutral peptide (CMSQAALNTRNSEEEVSSRRNNGTRHHHHHH) labelled with a single Alexa 488 fluorophore on its only Cys residue at the N-terminus to the EcPL membrane containing 0.1 mol% dioctadecylamine (DODA)-tris-Ni-NTA (*Beutel et al., 2014*).

### FRET-based in vitro PG synthesis assay in SLBs using TIRF microscopy

PG synthesis reactions were carried out at 10 mM Tris pH 7.5 supplemented with 1 mM MgCl$_2$, with or without 1 mM ampicillin and in the presence of 4 µM LpoB(sol). The reaction was started by addition of 4 µM of unlabelled lipid II. TIRF microscopy, using a set up described

elsewhere (*Baranova et al., 2020*), was used to monitor an increase in FRET efficiency and spatial reorganization of FRET signal over the time course of PG synthesis. To detect real-time FRET on supported lipid membranes, we used the so-called 'acceptor photobleaching approach' where a region of interest of about $10 \times 10$ μm was photobleached in the acceptor channel (lipid II-Atto647n) and the increase in fluorescence intensity of the donor (lipid II-Atto550) was recorded within a delay of 1 s. We found that in our experiments photobleaching of the acceptor dye was the only process that contributed to the recorded increase in the donor fluorescence signal. Accordingly, the relative increase in donor fluorescence can be used as a direct readout for the FRET efficiency and could therefore be calculated as described (*Loose et al., 2011*; *Verveer et al., 2006*). Briefly, donor intensity levels were calculated before ($I^D$) and after photobleaching ($I^{D,pb}$) using intensity measurements in ImageJ. FRET efficiency was calculated using *Equation 5*:

$$E = (I^{D,pb} - I^D)/I^{D,pb} \tag{5}$$

For time-course measurements (*Figure 4D*), the acceptor signal (lipid II-Atto647n) was photobleached every minute after the initiation of the reaction (the data point at time 0 corresponds to the addition of unlabelled lipid II). For each time point, a new region of interest in the same chamber was photobleached, and the change in the donor intensity was recorded to calculate FRET efficiency using *Equation 1*.

To have a control on the lipid membrane integrity during PG synthesis, the phospholipid DODA-tris-Ni-NTA (*Beutel et al., 2014*) was included during reconstitution at a 0.1 mol% ratio. DODA-tris-Ni-NTA was then visualized using a $His_6$-containing peptide (CMSQAALNTRNSEEEVSSRRNNG TRHHHHHH) labelled with Alexa488 on its single Cys residue, which we added in the same experiment in which we performed FRET analysis. To compare the fluidity and immobile fraction of lipid II-Atto647n before and after 1 hr of the synthesis reaction with the fluidity of phospholipids in the lipid membrane, the same region of interest was photobleached with a laser first at 640 nm and afterwards at 480 nm.

## In vitro PG synthesis assay using radiolabelled lipid II on SLBs

To assay PG synthesis on SLBs using radioactively labelled lipid II, we first reconstituted $PBP1B^{Ec}$ on SLBs containing EcPL extract and a $1:10^5$ $PBP1B^{Ec}$ to lipid molar ratio, as described above. Due to the low density of the enzyme, several $1.1$ $cm^2$ chambers were assayed for every condition in order to accumulate a measurable signal. In every chamber, reactions were started by addition of 10 μM [$^{14}$C]-labelled lipid II and 4 μM LpoB(sol) in a total volume of 100 μL per chamber. The synthesis reaction was carried out in 10 mM Tris pH 7.5, 1 mM $MgCl_2$. The chambers were incubated overnight (~16 hr) at 37°C and covered with parafilm. Reactions were stopped by addition of 100 μM moenomycin. To digest the produced PG, cellosyl was added at 0.05 g/L in the presence of 0.3% Triton X-100. After 1 hr incubation at 37°C, samples from six chambers were pooled in an Eppendorf tube, concentrated using a speed-vac evaporator, reduced using sodium borohydride, and analysed by HPLC as described above. For the experiment to determine lipid II incorporation and the localization of the produced PG, before addition of moenomycin, chambers were washed by removal of 50 μL of buffer and addition of 50 μL of fresh buffer while mixing. This was repeated five times. The removed volume from each wash was pooled and treated the same as the samples left in the chamber.

## Single-molecule tracking and analysis

To perform single-molecule tracking, MGC-$^{64}$PBP1B-his C77S C795S was labelled with the photostable far-red dye Dy647N as described above and then reconstituted into a polymer-supported lipid membrane as described elsewhere (*Roder et al., 2011*; *Roder et al., 2014*). Single-molecule tracking experiments were performed at a low protein-to-lipid molar ratio ($1:10^{-6}$). At this ratio, supported lipid membrane was largely homogeneous with the lowest immobile fraction from all the ratios tested (*Figure 3—figure supplement 1*). The single-molecule motion of PBP1B was measured prior to and after the addition of 1.5 μM lipid II after 15 min ex situ incubation in the presence of 10 mM HEPES pH 7.4, 150 mM NaCl, 1 mM $MgCl_2$ buffer and in the absence of LpoB(sol). The localization and tracking of PBP1B particles were performed by the SLIMfast software (*Roder et al., 2014*). To ensure that non-specifically stuck PBP1B particles did not contribute to the measured diffusion coefficient, the immobile particles were excluded using the DBSCAN spatial clustering algorithm

(*Sander et al., 1998*) with the following clustering parameters: a search area of 100 nm and a minimal time window of 30 frames at 65 ms/frame acquisition time. The displacement distribution for active PBP1B (in the presence of lipid II) was compared to the displacement distribution of PBP1B before lipid II addition by fitting the two-component Rayleigh distribution and comparing the weighted contribution of each population. The mean-squared displacement (MSD) was fitted to each individual trajectory longer than 650 ms (10 frames). Each MSD curve was fitted with a linear fit considering max 30% of the lag time for each trajectory.

## FRAP analysis

To control membrane fluidity upon the reconstitution of the transmembrane PBP1B (*Figure 3—figure supplement 1*, *Figure 4—figure supplement 1*) and fluidity of lipid II Atto-647n during PG synthesis (*Figure 4E, F*), we used a Matlab-based GUI frap_analysis (*Jönsson, 2020*) in details described elsewhere (*Jönsson et al., 2008*). This code allows to quantify the contribution of the immobile fraction to the estimated diffusion coefficient and is particularly suitable for the analysis of 2D diffusion with the photobleaching contribution during the recovery.

## Acknowledgements

We thank Alexander Egan (Newcastle University) for purified proteins LpoB(sol) and LpoP$^{Pa}$(sol), Federico Corona (Newcastle University) for purified MepM, and Oliver Birkholz and Jacob Piehler (Department of Biology and Center of Cellular Nanoanalytics, University of Osnabrück) for their help with PBP1B reconstitution into polymer-SLBs and initial guidance on single particle tracking. We also acknowledge Christian P Richter and Changjiang You (Department of Biology and Center of Cellular Nanoanalytics, University of Osnabrück) for providing SLIMfast software and tris-DODA-NTA reagent, respectively. This work was funded by the BBSRC grant BB/R017409/1 (to WV), the European Research Council through grant ERC-2015-StG-679239 (to ML), and long-term fellowships HFSP LT 000824/2016-L4 and EMBO ALTF 1163–2015 (to NB).

# Additional information

## Funding

| Funder | Grant reference number | Author |
|---|---|---|
| BBSRC | BB/R017409/1 | Waldemar Vollmer |
| European Research Council | ERC-2015-StG-679239 | Martin Loose |
| EMBO | EMBO ALTF 1163-2015 | Natalia Baranova |
| Human Frontiers Science Program | HFSP LT 000824/2016-L4 | Natalia Baranova |

The funders had no role in study design, data collection and interpretation, or the decision to submit the work for publication.

## Author contributions

Víctor M Hernández-Rocamora, Conceptualization, Data curation, Formal analysis, Validation, Investigation, Methodology, Writing - original draft, Writing - review and editing; Natalia Baranova, Conceptualization, Data curation, Formal analysis, Funding acquisition, Validation, Investigation, Visualization, Methodology, Writing - original draft, Writing - review and editing; Katharina Peters, Data curation, Formal analysis, Investigation, Writing - original draft, Writing - review and editing; Eefjan Breukink, Resources, Writing - review and editing; Martin Loose, Conceptualization, Data curation, Formal analysis, Funding acquisition, Validation, Visualization, Methodology, Writing - original draft, Project administration, Writing - review and editing; Waldemar Vollmer, Conceptualization, Data curation, Formal analysis, Supervision, Funding acquisition, Validation, Methodology, Writing - original draft, Project administration, Writing - review and editing

## Author ORCIDs

Víctor M Hernández-Rocamora (iD) https://orcid.org/0000-0003-2517-5707
Natalia Baranova (iD) https://orcid.org/0000-0002-3086-9124
Martin Loose (iD) https://orcid.org/0000-0001-7309-9724
Waldemar Vollmer (iD) https://orcid.org/0000-0003-0408-8567

## Decision letter and Author response

Decision letter https://doi.org/10.7554/eLife.61525.sa1
Author response https://doi.org/10.7554/eLife.61525.sa2

## Additional files

### Supplementary files

- Supplementary file 1. Table of oligonucleotides used in this study.
- Transparent reporting form

### Data availability

All data generated or analysed during this study are included in the manuscript and supporting files. Source data files have been provided for Figures 1–5 and the corresponding figure supplements.

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
