## [Decision Letter]

**Acceptance summary:**

In the work by Hernandez-Rocamora, a new FRET assay was developed to monitor the peptidoglycan synthesis of bifunctional PBPs. This new assay has the major advantages of being real-time, applicable to membrane-embedded enzymes, and able to monitor both GTasea and TPase activities at the same time. While there exist complications in the interpretation of the FRET signal quantitatively, with appropriate controls this assay will greatly help researchers in the field to determine and compare enzymatic activities and kinetics semi-quantitatively. The assay also has the potential to be developed further into single-molecule assays, allowing mechanistic studies of cell wall synthesis enzymes.

**Decision letter after peer review:**

Thank you for submitting your article "Real time monitoring of peptidoglycan synthesis by membrane-reconstituted penicillin binding proteins" for consideration by *eLife*. Your article has been reviewed by three peer reviewers, one of whom is a member of our Board of Reviewing Editors, and the evaluation has been overseen by Olga Boudker as the Senior Editor. The reviewers have opted to remain anonymous.

The reviewers have discussed the reviews with one another and the Reviewing Editor has drafted this decision to help you prepare a revised submission.

As the editors have judged that your manuscript is of interest, but as described below that the manuscript needs to be substantially revised to provide new biological insight if possible, with additional control experiments, critical analyses and clarifications.

We would like to draw your attention to changes in our revision policy that we have made in response to COVID-19 (https://elifesciences.org/articles/57162). First, because many researchers have temporarily lost access to the labs, we will give authors as much time as they need to submit revised manuscripts. We are also offering, if you choose, to post the manuscript to bioRxiv (if it is not already there) along with this decision letter and a formal designation that the manuscript is "in revision at *eLife*". Please let us know if you would like to pursue this option. (If your work is more suitable for medRxiv, you will need to post the preprint yourself, as the mechanisms for us to do so are still in development.)

Summary:

The manuscript by Hernández-Rocamora et al. describes a FRET-based real time assay that allows for continuous monitoring of peptidoglycan synthesis by aPBPs. The authors examine how the assay performs in detergent and in liposomes; they also assess whether the assay will be useful for studying regulators of peptidoglycan biosynthesis enzymes by applying it to previously studied lipoprotein activators of aPBPs, and a new lipoprotein activator that is similar to a previously studied one. The major advantage of the current assay in comparison to the original dansylated lipid II assay is that the FRET assay can monitor both GTase and TPase activities. Finally, the current FRET assay has the potential to become single-molecule study-compatible. The Vollmer lab is one of only a few that can carry out studies of PG biosynthetic enzymes, and he has teamed up with an expert in microscopy on this manuscript. The experiments are technically challenging and generally well done and the technical advances should be useful to the field. However, new biological/mechanistic insights are few. The work also needs to be significantly strengthened with more critically analyzed data, control experiments and better writing.

Reviewer #1:

1) FRET controls and calculations:

a) The authors should provide donor-only and acceptor-only controls under the same conditions for all experiments to allow the calculation of FRET efficiency quantitatively (preferably using the Ratio A method Clegg RM, Methods in Enzymology, 1992, 211, p353). This quantification is important because one major potential advantage of this method could be studying the kinetics of GTase and TPase activities, which may offer mechanistic insight on the enzyme's working. The Facceptor /Fdonor ratio measurement is convenient and can be useful for screening of a large sample size but is prone to artifacts such as fluorescence quenching due to reasons other than FRET, especially when the way the system is set up does not allow precise determination of the donor/acceptor ratio and attachment positions. These donor only or acceptor only experiments are also important for the FRAP measurement in the SLB assay.

b) In Figure 4 the authors used a ratiometric FRET equation Erel = IA/(IA+ID) in the semi-single-molecule assay to determine FRET efficiency. This measurement should be unified to allow direct comparison of FRET between different types of experiments (solubilized, liposomes, SLBs, bulk or single-molecule). As mentioned above, the acceptor intensity consists of the intensities from donor-excitation (FRET), direct-excitation, and bleed-through from the donor. Donor emission could be quenched due to reasons (such as the local environment as that in the dansyl-lipid II assay) other than FRET. More control experiments will allow the authors to determine how this impact the true acceptor signal resulting from FRET.

c) There appears to be a major difference in the fluorescence intensity spectra between the detergent-solubilized (Figure 1—figure supplement 1C) and liposome experiments (Figure 2—figure supplement 2A, C, D). In the detergent solubilized experiments, there appears to be little change in the intensity of the donor peak from PBP1B to +LpoB. However, in the liposome experiments the LpoB has a drastic loss of donor signal compared to that of PBP1B. This is another reason why calculating FRET efficiency instead of the relative ratio would allow for a better comparison between the experiments. This drastic loss of donor signal in the liposome experiments is seen in all 3 species. The authors should address why there is such a difference in the FRET profiles between these two experiments.

d) The authors rely on "Relative Slope" to show how the addition of LpoB impacts the PBP1B activity. Is the "Relative slope" calculated by normalizing the slope of a reaction with LopB to that without LopB? How is the time window determined? As this ratio does not really inform on the enzyme activity, it would be difficult for other groups to use the same method and to compare to this metric. It is also difficult to judge whether the GTase is faster or slower in the presence of TPase activity without the absolute values. Please provide the absolute measurement of the slope (or the initial velocity V0) using a better determined FRET efficiency value as in point 1a.

e) Additionally, perhaps the authors could determine the reaction velocities at different substrate concentrations to see if Michaelis-Menten kinetics could be applied? A reanalysis of the data with better quantifications with regard to enzyme's kinetics would benefit the community significantly by providing a more complete picture of enzyme's activity rather than only comparing PBP1B alone to +LpoB.

2) According to the authors' data and others' work, PBP1B can only synthesize short, uncrosslinked PG chains when its TPase is inhibited. When the TPase is not inhibited, the crosslinking could somehow help and accelerate the PG chains synthesis by the GTase. The entire signal change should be because of both GTase and TPase activities. The authors mentioned that the first slow phase reflects the GTase activity and the second fast phase reflects the TPase, but I do not find it convincing.

a) Please define which section is slow and which section is fast in the FRET curves-in the presence of LpoB, it does not appear to have a slow-phase if the initial lag phase of rise is the slow-phase the authors mentioned.

b) In the presence of Amp, the slope of the initial rise appears to be slower than that in the absence of Amp, which, could be the reason that the authors argue that TPase help accelerate GTase. Please provide absolute quantifications using FRET efficiency to show that this is indeed the case.

c) Can the authors add D,D-endopeptidase in the system after the end of the reaction to uncrosslink the peptides and see how much FRET signal remains, which should be contributed by the pure GTase activity? This signal could be compared with that in the presence of Amp to examine the authors' claim.

3) The lag of TPase activity behind GTase activity as shown in Figure 1F is intriguing. The lag appears to be on the order of 5-10 min in this system and suggest that the GTase and TPase activities are not synchronous. Does this mean that even though PBP1B has both activities in one single molecule, but the two reactions are catalyzed asynchronously by two different molecules?

4) The authors suggest that the decrease of FRET signal in Figure 2C, E and G is due to additional changes of fluorophore's spectroscopic properties. If so why such changes were not observed in the detergent or SLB experiments, if the product is the same? The authors should obtain parallel time points using radio-labeled Lipid II and run the products of both assays on HPLC to show that the decrease of FRET signal in these experiments did not correspond to changed product.

5) The SLB assay has great potentials to allow single-molecule measurement of glycan chain elongation and crosslinking kinetics. It is disappointing, though, to find out that no single-lipid II molecules in the polymerized PG can be visualized. Is this because the lipid II concentration too high on the SLB? Does it need to be that high?

6) To fully drive home the message that the new FRET is better in many ways than previous assays, the authors should compare it with the dansyl-lipid II assay side by side (i.e., not putting the other assay in the supplemental figures) and quantitatively.

7) Similarly, there appeared to be significant differences in FRET signals between the detergent, liposome and SLB assays with or without Amp, and it is difficult to compare the enhancement by LpoB among these assays. Please address these differences.

Reviewer #2:

1) It is stated that: "As expected, ampicillin did not prevent the stimulation of PBP1B by LpoB(sol), which accelerated the FRET increase by 10-20 times with or without ampicillin, consistent with the previously reported stimulation of both GTase and TPase activities." Although other results may suggest TPase activity is stimulated, one cannot draw the conclusion from the FRET assay results here that LpoB stimulates TPase activity – only that TPase activity boosts the FRET signal. There is some confusion in the manuscript about what can and cannot be concluded about mechanism from the FRET assays.

2) The authors use initial rates from the FRET assay to compare effects of LpoB in Figure 1 (and later do the same for other conditions/other activators). Initial rate conditions are used to quantify reaction rates for enzymes because under these conditions substrate depletion and product inhibition do not impact the rate. Initial rates are misleading when comparing a hyperbolic progress curve to a sigmoidal progress curve. The authors should comment on this, clarify the time frame over which the initial rates were measured (perhaps indicated with a figure), and also report the slopes of the "fast" phase of the reactions where activator is not present.

3) There is a pattern of low mass bands in the PAGE gel in Figure 1D for the LpoB reactions; these are barely visible in the reactions without LpoB. At the same time, the glycan strands are shorter in length for the LpoB reaction compared to the reaction without LpoB (lanes 2 and 6). (Shorter glycan strands were observed previously for LpoB - e.g, Paradis-Bleau et al.; Lupoli et al.). I am curious about whether these observations are connected (and therefore whether they are connected to how LpoB achieves activation).

4) The authors comment that the combined assays suggest that the total FRET signal emerges from two steps that have different rates. I recommend either refining or removing this comment. The statement suggests that it might be possible to extract rates for the two steps alluded to from the FRET signal. But there are more than two steps and two rates. For polymerization reactions, initiation and elongation rates are often substantially different and there is evidence in the literature that this is case for PG synthesis. (The slow rate of reaction for the "only labelled Lipid II" reactions does not report on the rate of polymerization with unlabeled Lipid II present because the products are very short – so the polymerization does not "take off" the way it does in the other reactions.) Moreover, there are likely two rates for crosslinking in vitro because the first crosslinks are formed intermolecularly and later crosslinks are formed intramolecularly. Finally, the FRET signal itself is different for uncrosslinked and crosslinked PG – as the authors note, it reaches a higher maximum for the latter.

5) Figure 2C: was the lane 7 sample in the gel the same as the one for which the fluorescence trace is shown? The fluorescence trace looks flat but lane 7 shows polymerization. In other gels, other lanes that show comparable levels of polymerization also show a FRET signal.

6) I already mentioned that the discussion could use some work, but wanted to elaborate. The explanation for how TPase activity increases the FRET signal is plausible and supported by the data. But the section on "coupled reactions in Class A PBPs" is confusing because it is not clear what the authors are trying to accomplish with it. There has been a lot of debate about what LpoB does to activate PBP1b and how it does it. That LpoB affects both GTase activity and TPase activity has been shown repeatedly to one degree or another, and the idea that one activity influences the other in this context has been discussed explicitly going back at least to 2014 (Egan et al., 2014; Lupoli et al., 2014). The authors discuss the idea again here, evidently in response to a recent paper (Catherwood et al.) that attributes LpoB's essentiality to its effects on TPase activity. They conclude that Catherwood's conditions may have overemphasized the TPase effect. I don't think arguing this point is the most important use of the discussion space. The authors should clarify how they think about what the FRET assays show with respect to mechanism of Lpo activators and try to connect their work to a path forward.

Summary: there is a lot of nice material in this manuscript and I believe the authors are pushing the field ahead with the FRET assays, particularly their use in SLMs. However, the analysis of the results is not at the same level as the technical innovations and this should be improved. Mainly text revisions could make the paper suitable for *eLife*, but it will take some work on the part of the authors.

Reviewer #3:

This manuscript describes studies of generation of improved in vitro assay systems for peptidoglycan synthesis, using liposomes and supported lipid bilayers. This work can be very influential as this synthetic process is a primer candidate for the development for additional antibiotics, and the detection methods utilized are potentially adaptable to high-throughput screens.

Overall the paper is well written and the data are convincing. I have a few comments below, primarily concerning the data presentation.

1) Figures 1A, 2B, and 5: These figures show some of the fluorescently-labeled peptides as being involved in crosslinks. I do not think there is evidence in this paper that these substrates are ever used by transpeptidases. They do not appear to show up in obvious radioactively-labeled HPLC peaks. Doesn't this suggest that they are NOT being crosslinked? All off the crosslinked material seen in HPLC peaks is simply the radiolabeled Lipid II.

2) Figure 1C: I don't understand why the slope with LpoB is greater in the presence of Amp? Can the authors comment?

[Editors' note: further revisions were suggested prior to acceptance, as described below.]

Thank you for submitting your article "Real time monitoring of peptidoglycan synthesis by membrane-reconstituted penicillin binding proteins" for consideration by *eLife*. Your article has been reviewed by three peer reviewers, one of whom is a member of our Board of Reviewing Editors, and the evaluation has been overseen by Olga Boudker as the Senior Editor.

The reviewers have discussed the reviews with one another and the Reviewing Editor has drafted this decision to help you prepare a revised submission.

As the editors have judged that your manuscript is of interest, but as described below that additional experiments are still required before it is published. We would like to draw your attention to changes in our revision policy that we have made in response to COVID-19 (https://elifesciences.org/articles/57162). First, because many researchers have temporarily lost access to the labs, we will give authors as much time as they need to submit revised manuscripts. We are also offering, if you choose, to post the manuscript to bioRxiv (if it is not already there) along with this decision letter and a formal designation that the manuscript is "in revision at *eLife*". Please let us know if you would like to pursue this option. (If your work is more suitable for medRxiv, you will need to post the preprint yourself, as the mechanisms for us to do so are still in development.)

Major comments:

The major advantages of the assay are three-fold, (1) it is a real-time; (2) it is applicable to membrane-embedded enzymes, and (3) it can monitor both GTase and TPase. The revised manuscript has provided evidence for all the three with improved clarity, new control experiments, and better FRET efficiency calculations, but it still lacks the rigor in the quantitative (or semi-quantitative) aspect of the assay. Specifically, the authors need to provide evidence that the F_acceptor_/F_donor_ ratio indeed reflects proportionally to either just the GTase activity or a combined GTase and TPase activity. The authors did show qualitatively that this ratio is altered when the activator LpoB (or P) or drugs are used, but it is difficult to use this ratio to compare quantitatively (or semi-quantitatively) the rates of different reactions catalyzed by different enzymes, or the same enzyme but with different mutations and regulators. Being able to do so will greatly enhance the impact of the FRET assay and promote its wide adoption in the field.

It appears that the measured F_acceptor_/F_donor_ ratio is not only dependent on the GTase and TPase activities of PBP1B, but some other effects that influence the fluorescence of the dyes. These other effects will make it difficult to use the FRET assay to quantitatively compare different reactions. For example, in Figure 3C, the reaction with Amp (VII) barely had any signal change in F_acceptor_/F_donor_, but the corresponding gel lane cleared showed polymerized, linear PG strands, suggesting that the polymerization reaction is not reflected in the fluorescence ratio measurement. The rise and decay of the +LpoB/Amp (III) reaction also look strange-this reaction had a similar PG ladder on the gel as reaction (-LpoB/Amp)VII, but the fluorescence ratio measurements are drastically different. Another example is the comparison between the three PBP1Bs: it appears that crosslinking activities have different contributions to the fluorescence ratio measurement: ampicillin completely abolished the fluorescence ratio signal for PBP1BEc but not for PBP1BAb. It is hard to imagine that these highly homologous enzymes would have very different mechanisms to synthesize PG.

A likely explanation for these different fluorescence signals in these reactions is probably what the authors suggested, that some spectroscopic properties of the dyes changed due to the incorporation and/or local environment. Another possibility is that these enzymes of different species may handle the dye-labeled LipidII differentially, one PBP1B may have more difficulty in accepting the dye-labeled LipidII than the other PBP1B. Yet another possibility could also be that the stimulation effect of LpoB is simply that now more dye-labeled LipidII v.s. unlabeled Lipid II can be incorporated, but not necessarily the rate at which Lipid II is incorporated.

It is understandable that these in vitro experiments are extremely challenging, but fluorescence measurements are sensitive and prone to artifacts, and the authors should perform control experiments to make sure that the measured signal indeed reflects the desired enzymatic activities. In particular, in calculating the fluorescence ratio, the donor-only and acceptor-only controls need to be run simultaneously to determine how much fluorescence change is due to the local environment change, and how much is due to FRET. The authors have performed these control experiments but only used them for the end-point measurements. These controls should also be performed for the real-time measurements (with LpoB, with or without Amp), which could help to explain some strange changes in the measured fluorescence ratio (such as that in Figure 3C).

The goal, hopefully, is that one can confidently interpret the ~ 25-fold higher initial slope of PBP1BEc with LpoB in the presence of ampicillin as a 25-fold (or proportionally) higher GTase activity of PBP1BEc stimulated by LpoB (such as that in Figure 3D). Similarly, one can use the smaller fold-increase of the initial slopes in the cases of PBP1BAb and PBP1BPa in the presence of their cognate activators to draw the conclusion that these activators do not stimulate these PBP1Bs to the same extent as that of PBP1BEc. If the fold enhancement of the initial slopes by LpoB in the presence and absence of ampicillin are similar to each other, one perhaps can say that here LpoB only stimulates the GTase activity, and so on and so forth.

The new FRET assay bears great potential in providing a sensitive and quantitative way to characterize both GTase and TPase activities in ways not possible before. The Vollmer laboratory also has the expertise to make this assay hugely impactful for the field. These experiments can become powerful, but they need critical controls.

---

## [Author Response]

Summary:The manuscript by Hernández-Rocamora et al. describes a FRET-based real time assay that allows for continuous monitoring of peptidoglycan synthesis by aPBPs. The authors examine how the assay performs in detergent and in liposomes; they also assess whether the assay will be useful for studying regulators of peptidoglycan biosynthesis enzymes by applying it to previously studied lipoprotein activators of aPBPs, and a new lipoprotein activator that is similar to a previously studied one. The major advantage of the current assay in comparison to the original dansylated lipid II assay is that the FRET assay can monitor both GTase and TPase activities. Finally, the current FRET assay has the potential to become single-molecule study-compatible. The Vollmer lab is one of only a few that can carry out studies of PG biosynthetic enzymes, and he has teamed up with an expert in microscopy on this manuscript. The experiments are technically challenging and generally well done and the technical advances should be useful to the field. However, new biological/mechanistic insights are few. The work also needs to be significantly strengthened with more critically analyzed data, control experiments and better writing.

We thank the reviewers for their positive assessment of our manuscript and acknowledging the usefulness of the assay. We would like to stress that, in addition to being able to monitor both reactions, the other major advantage of the new assay is that it is capable of measure PG synthesis with membrane-reconstituted enzymes. The dansylated lipid II assay does not work with liposomes or supported bilayers. We are also grateful for the insightful points made by the reviewer. Based on these points we critically analyzed the data, added the suggested control experiments and improved the writing of the manuscript. We hope that the reviewers agree with us that these changes have significantly improved the manuscript.

Reviewer #1:1) FRET controls and calculations:a) The authors should provide donor-only and acceptor-only controls under the same conditions for all experiments to allow the calculation of FRET efficiency quantitatively (preferably using the Ratio A method Clegg RM, Methods in Enzymology, 1992, 211, p353). This quantification is important because one major potential advantage of this method could be studying the kinetics of GTase and TPase activities, which may offer mechanistic insight on the enzyme's working. The Facceptor /Fdonor ratio measurement is convenient and can be useful for screening of a large sample size but is prone to artifacts such as fluorescence quenching due to reasons other than FRET, especially when the way the system is set up does not allow precise determination of the donor/acceptor ratio and attachment positions. These donor only or acceptor only experiments are also important for the FRAP measurement in the SLB assay.

The reviewer is correct in that we did not provide donor-only and acceptor-only controls to calculate FRET efficiency quantitatively. To address this point, we have now measured apparent FRET efficiencies using the suggested ratio A method for the end products of the reactions of *E. coli* PBP1B in detergents in the presence ampicillin or moenomycin, or without antibiotic. We measured samples containing only lipid II-Atto647n+lipid II (acceptor reference) and samples containing only lipidII-Atto550+lipid II (donor reference). With these data we demonstrate that signals arising in these reactions are indeed due to FRET, and we provide the values for FRET efficiencies. The new data shown in Figure 1B-D.

We respectfully question the reviewer suggestion to calculate FRET efficiencies when monitoring reactions in real time, for several reasons. First, FRET efficiencies will not be useful to provide any meaningful distances between fluorophores, because probe molecules are unlikely to be positioned homogeneously within the PG produced. Very likely, probe molecules are separated by variable distances and with a range of different orientations, giving rise to a mixture of FRET between probes from the same glycan chain and different chains.

Second, calculating FRET efficiency in real time would require combining signals arising from different reactions in different wells (the references with donor-only or acceptor-only plus the sample with both) which, in practice, would substantially increase the noise in the final reaction curve. This problem would be exacerbated in the membrane system, which would require to use different liposome preparations for the different samples (because the labelled lipid versions need to be reconstituted together with the enzyme).

We believe that the F_acceptor_/F_donor_ ratio is useful even if a small part of the signal might come from quenching. We demonstrated that the F_acceptor_/F_donor_ ratio monitors PG synthesis reactions, because the rise in the ratio requires the incorporation of the probes into peptidoglycan, as verified by the controls with ampicillin and moenomycin.

To measure FRET within a PG layer on a supported lipid bilayer, we performed acceptor photobleaching or donor dequenching experiments, where the energy transfer from the donor to the acceptor is irreversibly blocked and its intensity is increased. The relative increase in donor fluorescence can be used a direct readout for the FRET efficiency as photobleaching of the acceptor dye is the only process that contributes to the recorded increase in the donor fluorescence signal (see for example Verveer et al., 2006). In rare cases, an increase in donor fluorescence can be recorded even in the absence of acceptor dye, which could result in an overestimation of the FRET efficiency. To our knowledge, this has so far only been found when fluorescent proteins were used (Karpova et al., 2003). Additionally, the donor signal could appear brighter due to spectral crosstalk. In this the FRET signal would be underestimated as the relative intensity increase after acceptor bleaching is decreased.

To test for these two opposing effects, we performed several control experiments. First, in the presence of Ampicillin, there was no increase in donor intensity and the calculated FRET efficiency was 0 (Figure 5B). Second, the signal was dramatically reduced after treatment with cellosyl and when the PBP1B E233Q mutant was used (Figure 5B-C). Additionally, we now also performed a control experiment, where only Lipid II Atto550 and Atto647n, and not PBP1B, are present in the assay (see Author response image 1). Also under these conditions, the donor intensity was unchanged upon acceptor bleaching and the calculated FRET efficiency was therefore 0. We therefore conclude that there is no contribution of acceptor fluorescence in the donor channel in the absence of PG synthesis.

Together with the controls performed in our spectroscopic assays in solution, we are confident that the increase in donor intensity was soley due to the activity of PBP1B and not caused by an unidentified photophysical process or spectral crosstalk. We added a comment on this issue to the revised text.

**Author response image 1. sa2fig1:** Only Lipid II-Atto647N and Lipid-Atto550 were present in the membrane. After photobleaching of the acceptor dye (Att647N), we did not observe an donor increase in intensity in the bleached area.

b) In Figure 4 the authors used a ratiometric FRET equation Erel = IA/(IA+ID) in the semi-single-molecule assay to determine FRET efficiency. This measurement should be unified to allow direct comparison of FRET between different types of experiments (solubilized, liposomes, SLBs, bulk or single-molecule). As mentioned above, the acceptor intensity consists of the intensities from donor-excitation (FRET), direct-excitation, and bleed-through from the donor. Donor emission could be quenched due to reasons (such as the local environment as that in the dansyl-lipid II assay) other than FRET. More control experiments will allow the authors to determine how this impact the true acceptor signal resulting from FRET.

The reviewer is correct that a different equation was used to calculate FRET efficiency 4 than for detergents or liposomes. However, the equation we used is not the one indicated by the reviewer but:

(1) E = (ID,pb – ID) / ID,pb = 1 – (ID/ID,pb)

Where ID,pb is the donor intensity before photobleaching and ID,pb is the donor intensity after photobleaching (Verveer et al., 2006). This equation uses the donor fluorescence to calculate FRET, the acceptor fluorescence is not relevant in this calculation.

Due to the very low concentration and density of PBP enzyme on the supported bilayer the comparison of reaction rates from these experiments with the ones from liposomes or detergents is not meaningful. For a description of our controls performed, please refer to our answer to question 1a

c) There appears to be a major difference in the fluorescence intensity spectra between the detergent-solubilized (Figure 1—figure supplement 1C) and liposome experiments (Figure 2—figure supplement 2A, C, D). In the detergent solubilized experiments, there appears to be little change in the intensity of the donor peak from PBP1B to +LpoB. However, in the liposome experiments the LpoB has a drastic loss of donor signal compared to that of PBP1B. This is another reason why calculating FRET efficiency instead of the relative ratio would allow for a better comparison between the experiments. This drastic loss of donor signal in the liposome experiments is seen in all 3 species. The authors should address why there is such a difference in the FRET profiles between these two experiments.

The activity of PBP1B without LpoB is lower in liposomes than in detergents, hence the difference is more prominent in the end-point spectra of PBP1B versus PBP1B+LpoB in liposomes. We now discuss this difference in the revised text. We also added further comments on the difference between the behaviour of PBPs in the membrane and in detergents to Discussion.

d) The authors rely on "Relative Slope" to show how the addition of LpoB impacts the PBP1B activity. Is the "Relative slope" calculated by normalizing the slope of a reaction with LopB to that without LopB? How is the time window determined? As this ratio does not really inform on the enzyme activity, it would be difficult for other groups to use the same method and to compare to this metric. It is also difficult to judge whether the GTase is faster or slower in the presence of TPase activity without the absolute values. Please provide the absolute measurement of the slope (or the initial velocity V0) using a better determined FRET efficiency value as in point 1a.

We thank the reviewer for making these points. The relative slope is indeed the ratio between the slope in the presence of LpoB and the slope without LpoB under the same conditions. This method has been used when measuring the activation by LpoB using the dansyl-Lipid II method to measure GTase activity (e.g. Egan et al., 2014, 2018) and we used this method in order to compare our results with those from the literature. We relabelled the plots from “Relative slope” to “Fold change in initial slope” to better describe them.

The slopes were calculated in the linear part of the curve after the signal started to increase. We now explain in the main text how slopes were calculated and provide more details in the experimental section. The absolute values for the slopes and the time window used to calculated them for each condition are now listed in the source files for the corresponding figures (Figures 1, 3, and Figure 3—figure supplement 5, Figure 3—figure supplement 6 and Figure 3—figure supplement 7).

e) Additionally, perhaps the authors could determine the reaction velocities at different substrate concentrations to see if Michaelis-Menten kinetics could be applied? A reanalysis of the data with better quantifications with regard to enzyme's kinetics would benefit the community significantly by providing a more complete picture of enzyme's activity rather than only comparing PBP1B alone to +LpoB.

To determine enzyme kinetics parameters was not our goal when developing this method although, in theory, the method should be useful for this purpose. Searching the literature we noticed that the kinetics of class A-PBP GTase and TPase activities have only been quantified separately, never both reactions together. We believe that developing a precise quantitative theoretical model for both activities is out of the scope of this paper and would require extensive new experiments and modelling.

We also noticed that the kinetics of the individual activities of *E. coli* PBP1B have been reported in the literature, but only for the detergent solubilized enzyme. Schwartz et al. (2002) established that glycosyltransferase activity followed Michaelis-Menten kinetics in the presence of the detergent decyl-PEG using an HPLC assay (Schwartz et al., 2002). Catherwood et al. (2020) used a D-Ala release assay to determined that PBP1B transpeptidase did not follow Michaelis-Menten in reactions with lipid II (mDAP) – same as here – and in the presence of Triton X-100. We now discuss these issues.

2) According to the authors' data and others' work, PBP1B can only synthesize short, uncrosslinked PG chains when its TPase is inhibited. When the TPase is not inhibited, the crosslinking could somehow help and accelerate the PG chains synthesis by the GTase. The entire signal change should be because of both GTase and TPase activities. The authors mentioned that the first slow phase reflects the GTase activity and the second fast phase reflects the TPase, but I do not find it convincing.

We would like to respectfully correct the statement made by the reviewer in the first sentence. TPase-inhibited PBP1B can indeed synthesise long chains (e.g., lane 2 on Figure 2C). The reviewer might have confused this fact with the results of control samples performed with only fluorescently-derivatised lipid II analogues (right panel in Figure 1E and lanes 4 and 8 in Figure 1—figure supplement 3B). We discuss that the binding of two labelled lipid II molecules to the two binding sites (donor and acceptor) in the GTase domain of PBP1B is probably hindered due to the big size of the Atto550 and Atto647n probes. Hence, in the absence of unlabelled lipid II, PBP1B on its own is unable to form glycan chains, but with LpoB PBP1B is stimulated to produce short glycan chains under these conditions.

We have now removed the statement cited by the reviewer, which was confusing and only intended for PBP1B reactions without LpoB. We did not mean to imply that all FRET signal was due to transpeptidation and only stated that transpeptidation was necessary for a substantial increase in FRET signal. Our model is that FRET arises from glycosyltransferase reactions (intra-chain FRET) and transpeptidation (inter-chain FRET), and the second contribution is higher in the absence of LpoB.

a) Please define which section is slow and which section is fast in the FRET curves-in the presence of LpoB, it does not appear to have a slow-phase if the initial lag phase of rise is the slow-phase the authors mentioned.

As the reviewer comments, there is a lag phase in the increase in FRET in reactions with only PBP1B, but this disappears in the presence of activator, LpoB (Figure 1E, left panel). This lag phase occurs in reactions with or without ampicillin (Figure 1E, middle panel) and it has been previously observed with the dansyl-lipid II method when measuring glycosyltransferase rates (Schwartz et al., 2002). The suppression of the lag phase by LpoB was also observed using the dansyl-lipid II method (Egan et al., 2014). Thus we hypothesize that LpoB accelerates the initiation of GTase, in addition to the rate of glycan chain elongation. In our case, we also have to take into account that LpoB allows PBP1B to incorporate two labelled lipid II molecules consecutively, as demonstrated in the controls with only Atto550- and Atto647n-labelled lipid II (Figure 1E, right panel). Thus, LpoB most likely enhances the intra-chain FRET, consistent with the observation that LpoB increased not only the reaction slope but also the final FRET signal in reactions with ampicillin when only intra-chain FRET is possible.

b) In the presence of Amp, the slope of the initial rise appears to be slower than that in the absence of Amp, which, could be the reason that the authors argue that TPase help accelerate GTase. Please provide absolute quantifications using FRET efficiency to show that this is indeed the case.

Sorry for the misunderstanding. We did not intend to claim that the TPase helps accelerate the GTase. In the statement we only meant to say that the FRET signal increased rapidly once transpeptidation occur in reactions without activator and without ampicillin. We therefore removed the confusing statement. We address the last sentence made by the reviewer in our reply to the next point.

c) Can the authors add D,D-endopeptidase in the system after the end of the reaction to uncrosslink the peptides and see how much FRET signal remains, which should be contributed by the pure GTase activity? This signal could be compared with that in the presence of Amp to examine the authors' claim.

Again, we did not intend to claim that the TPase helps accelerate the GTase. Our model was and still is that the in the absence of activator, inter-chain FRET is a major contributor to the FRET signal. This is now demonstrated in new data we present in the new Figure 1B-D in which we show that the addition of ampicillin (which blocks cross-link formation) causes a decrease of FRET efficiency to 30% of the value without antibiotic, and this result is commented in the revised text.

In addition, we carried out the experiment with a DD-endopeptidase, as suggested by the reviewer, and shown these new data in the new Figure 2A-B. We used the endopeptidase MepM to digest PG produced in the presence or absence of ampicillin. We found that MepM was unable to digest all cross-links in Atto-labelled PG, but caused a substantial reduction in the cross-linked PG (compare lanes M in the presence or absence of ampicillin on new Figure 2A). Consistent with this result, MepM digestion of the labelled PG reduced the FRET efficiency by 2-fold while, as expected, MepM did not significantly affect the FRET efficiency of the uncross-linked PG synthesised in the presence of ampicillin (new Figure 2D).

3) The lag of TPase activity behind GTase activity as shown in Figure 1F is intriguing. The lag appears to be on the order of 5-10 min in this system and suggest that the GTase and TPase activities are not synchronous. Does this mean that even though PBP1B has both activities in one single molecule, but the two reactions are catalyzed asynchronously by two different molecules?

Our results or published data do not suggest that both activities are catalysed “asynchronously”. Rather, PBP1B requires ongoing GTase reactions for efficient TPase activity (Bertsche et al., 2005) and Catherwood et al. (2020) recently demonstrated that PBP1B uses peptides in the growing glycan chain synthesised by the GTase domain as TPase donors (*i.e*. they link the D-Ala in fourth position in the pentapeptides of the growing chain to the mDAP group in an acceptor chain) (Catherwood et al., 2020).

One might expect a short lag between the start of the GTase and the beginning of the TPase because the growing chain has to grow at least to ~8 subunits to reach the TPase site (Sung et al., 2009). However, the experiment of Figure 1F (new Figure 2E) is not suited to measure a possible lag phase because of the sampling rate (every 5 min) and the possibility that, at the beginning of the reaction, small peaks are missed because of the detection limit and the inherent noise of radioactivity detection in chromatography. For example, in Figure 1F (new Figure 2E) at 5 min, the area of the Penta peak is only 459 CPM (over a peak width of 0.8 min) and a possible smaller TetraPenta peak would be barely detectable over the signal noise, which is ~90 CPM/min.

4) The authors suggest that the decrease of FRET signal in Figure 2C, E and G is due to additional changes of fluorophore's spectroscopic properties. If so why such changes were not observed in the detergent or SLB experiments, if the product is the same? The authors should obtain parallel time points using radio-labeled Lipid II and run the products of both assays on HPLC to show that the decrease of FRET signal in these experiments did not correspond to changed product.

Sorry, we are confused by the reviewer's question. We only see a downwards slope in the FRET curve of PBP1B reactions with LpoB and ampicillin in liposomes (old Figure 2C, middle panel), not in old Figure 2E or Figure 2G. We speculate in the text that this decrease could be due to the influence of the membrane on the fluorescence of the probe. If our speculation is correct, this decrease should not be observed on SLBs because due to the very low lipid II concentrations in those experiments, little PG is produced and, as inter-chain FRET is the major contributor to the FRET signal, very little or no FRET signal is observed with ampicillin. In assays with detergents there is no membrane that can affect fluorescence properties and, consistent with our speculation, we did not observe a negative slope (new Figure 1E, middle panel).

5) The SLB assay has great potentials to allow single-molecule measurement of glycan chain elongation and crosslinking kinetics. It is disappointing, though, to find out that no single-lipid II molecules in the polymerized PG can be visualized. Is this because the lipid II concentration too high on the SLB? Does it need to be that high?

The reviewer is correct that lipid II concentrations are too high in SLB for single molecule analysis, and this is so because concentrations low enough for single molecules do not support PG synthesis. We agree with the reviewer that single-molecule measurements of glycan chain elongation and cross-link formation is a worthy goal. We are working towards this goal but it is still a big step from where we are now with the current technology, which we believe already provides a major advance.

6) To fully drive home the message that the new FRET is better in many ways than previous assays, the authors should compare it with the dansyl-lipid II assay side by side (i.e., not putting the other assay in the supplemental figures) and quantitatively.

Both EcPBP1B and PaPBP1B have been studied by dansyl-lipid II assay in the literature (Schwartz et al., 2002, Egan et al., 2014, Caveney et al., 2020) and we did not repeat those published assays here. We performed the dansyl-lipid II assay for AbPBP1B and included it into the ms, because there was no prior information about this enzyme in the literature. We did not mean to claim that our assay is superior to the dansyl-lipid II because it provides better data, but because it allows to monitor transpeptidation and glycosyltransferase activity of the membrane-reconstituted enzymes (as is stated in the Discussion). Where relevant, we compared our results from the FRET assay with prior results in the literature obtained with dansyl-lipid II, *i.e.* when comparing the activation of the GTase rate by LpoB and LpoP regulators (“Coupled reactions in class A PBPs” section in the Discussion).

7) Similarly, there appeared to be significant differences in FRET signals between the detergent, liposome and SLB assays with or without Amp, and it is difficult to compare the enhancement by LpoB among these assays. Please address these differences.

The conditions in SLB are not comparable to the conditions in liposomes and detergents, due to the low density of PBP1B in SLBs. This difference is discussed in the text (Results section on SLBs). As for the comparison between liposomes and detergents, we may expect differences in behaviour of enzymes because of the following reasons: (1) possible effects of detergents on enzyme activity and (2) kinetics of enzymatic reactions occurring on the membrane are different from reactions occurring in solution. We comment about the specific difference in behaviour of the enzymes in membranes in the Discussion.

Reviewer #2:1) It is stated that: "As expected, ampicillin did not prevent the stimulation of PBP1B by LpoB(sol), which accelerated the FRET increase by 10-20 times with or without ampicillin, consistent with the previously reported stimulation of both GTase and TPase activities." Although other results may suggest TPase activity is stimulated, one cannot draw the conclusion from the FRET assay results here that LpoB stimulates TPase activity – only that TPase activity boosts the FRET signal. There is some confusion in the manuscript about what can and cannot be concluded about mechanism from the FRET assays.

We thank the reviewer for this point. We agree that the FRET assay only does not allow the conclusion about TPase rate. Some activators enable PBPs to polymerise Atto-lipid II without unlabelled lipid II (thus reducing the distance between probes within the chains and increasing intra-chain FRET), hence we cannot conclude from FRET results alone that the activators stimulate the TPase. We have carefully edited the text to avoid this inaccurate statement.

2) The authors use initial rates from the FRET assay to compare effects of LpoB in Figure 1 (and later do the same for other conditions/other activators). Initial rate conditions are used to quantify reaction rates for enzymes because under these conditions substrate depletion and product inhibition do not impact the rate. Initial rates are misleading when comparing a hyperbolic progress curve to a sigmoidal progress curve. The authors should comment on this, clarify the time frame over which the initial rates were measured (perhaps indicated with a figure), and also report the slopes of the "fast" phase of the reactions where activator is not present.

The slopes were calculated from the linear part of the curve when the signal started to increase after the lag phase (when present). For example, for reactions with detergent-solubilised *E. coli* PBP1B, we measured the slope from 10-20 min without LpoB (after the lag phase), but from 0-1.3 min with LpoB (no lag). The calculation of initial rates is now explained briefly in the Results text and in detail in the Materials and methods section. We now provide the absolute values for the slopes and the time windows used to calculate them for each condition in the source files for the corresponding figures (new Figures 1, 3, and Figure 3—figure supplement 5, Figure 3—figure supplement 6 and Figure 3—figure supplement 7). We have also relabelled the figures as “Fold change in initial rate” to avoid confusion.

3) There is a pattern of low mass bands in the PAGE gel in Figure 1D for the LpoB reactions; these are barely visible in the reactions without LpoB. At the same time, the glycan strands are shorter in length for the LpoB reaction compared to the reaction without LpoB (lanes 2 and 6). (Shorter glycan strands were observed previously for LpoB - e.g, Paradis-Bleau et al.; Lupoli et al.). I am curious about whether these observations are connected (and therefore whether they are connected to how LpoB achieves activation).

We thank the reviewer for pointing to this observation. We would like to be hesitant to use the relatively crude method of SDS-PAGE to compare smaller differences in glycan chain lengths, as they might occur between samples with or without LpoB. Egan *et al.* (2018) observed an increase in the heterogeneity of chain lengths produced by PBP1B in the presence of LpoB, contradicting the results by Paradis-Bleu *et al.* and Lupoli *et al.* (Egan et al., 2018). Egan et al. also provided evidence that the solvent demethyl sulphoxide (DMSO) used in high concentration in the other studies, reduced the PBP1B-LpoB affinity and might itself has an effect on glycan chain length distribution. We believe it is outside the scope of the current manuscript to clarify this issue.

4) The authors comment that the combined assays suggest that the total FRET signal emerges from two steps that have different rates. I recommend either refining or removing this comment. The statement suggests that it might be possible to extract rates for the two steps alluded to from the FRET signal. But there are more than two steps and two rates. For polymerization reactions, initiation and elongation rates are often substantially different and there is evidence in the literature that this is case for PG synthesis. (The slow rate of reaction for the "only labelled Lipid II" reactions does not report on the rate of polymerization with unlabeled Lipid II present because the products are very short – so the polymerization does not "take off" the way it does in the other reactions.) Moreover, there are likely two rates for crosslinking in vitro because the first crosslinks are formed intermolecularly and later crosslinks are formed intramolecularly. Finally, the FRET signal itself is different for uncrosslinked and crosslinked PG – as the authors note, it reaches a higher maximum for the latter.

We thank the reviewer for this point. We agree that our statement was confusing and we removed it from the manuscript. We thank the reviewer also for their comments on the different rates for GTase and TPase. We have added a more extensive discussion on GTase and TPase rates of class A PBPs, citing the relevant literature, and commenting on what can be concluded from our data on this topic in the rewritten “Coupled reactions in class A PBPs and their activation” section of the Discussion.

5) Figure 2C: was the lane 7 sample in the gel the same as the one for which the fluorescence trace is shown? The fluorescence trace looks flat but lane 7 shows polymerization. In other gels, other lanes that show comparable levels of polymerization also show a FRET signal.

We double checked the gels and the lanes are the same. Although there are some short glycan chains on lane 7, most of the lipid II remained unpolymerised. Besides, for FRET to occur in the absence of crosslinking more than one label has to be incorporated per chain and they should be sufficiently close together. The very low FRET thus indicates that very few probes are incorporated per chain at distances close enough to produce FRET.

6) I already mentioned that the discussion could use some work, but wanted to elaborate. The explanation for how TPase activity increases the FRET signal is plausible and supported by the data. But the section on "coupled reactions in Class A PBPs" is confusing because it is not clear what the authors are trying to accomplish with it. There has been a lot of debate about what LpoB does to activate PBP1b and how it does it. That LpoB affects both GTase activity and TPase activity has been shown repeatedly to one degree or another, and the idea that one activity influences the other in this context has been discussed explicitly going back at least to 2014 (Egan et al., 2014; Lupoli et al., 2014). The authors discuss the idea again here, evidently in response to a recent paper (Catherwood et al.) that attributes LpoB's essentiality to its effects on TPase activity. They conclude that Catherwood's conditions may have overemphasized the TPase effect. I don't think arguing this point is the most important use of the discussion space. The authors should clarify how they think about what the FRET assays show with respect to mechanism of Lpo activators and try to connect their work to a path forward.

We thank the reviewer for these points. Based on the reviewer's suggestions we have extensively rewritten the part about “Coupled reactions in Class A PBPs” in the Discussion. We now discuss the effects of Lpo activators on activities of PBPs more broadly, with an emphasis to what our results can add to the ongoing discussions. In particular, we discuss the importance of assaying PBPs in the membrane environment based on our findings that LpoB is required for the activities of several PBPs we studied, when they were reconstituted into the membrane. We also discuss that this observation agrees with the absolute requirement of LpoB for the function of PBP1B in the cell.

Reviewer #3:This manuscript describes studies of generation of improved in vitro assay systems for peptidoglycan synthesis, using liposomes and supported lipid bilayers. This work can be very influential as this synthetic process is a primer candidate for the development for additional antibiotics, and the detection methods utilized are potentially adaptable to high-throughput screens.Overall the paper is well written and the data are convincing. I have a few comments below, primarily concerning the data presentation.1) Figures 1A, 2B, and 5: These figures show some of the fluorescently-labeled peptides as being involved in crosslinks. I do not think there is evidence in this paper that these substrates are ever used by transpeptidases. They do not appear to show up in obvious radioactively-labeled HPLC peaks. Doesn't this suggest that they are NOT being crosslinked? All off the crosslinked material seen in HPLC peaks is simply the radiolabeled Lipid II.

We thank the reviewer for this point. Although it is technically possible that fluorescently-labelled pentapeptides participate as donors in the transpeptidase reaction, as depicted in our schemes, we found no evidence for this to happen in the HPLC assay. Thus, we have modified the schemes in the cited figures and now avoid showing Atto-labelled pentapeptides participating directly in cross-links.

2) Figure 1C: I don't understand why the slope with LpoB is greater in the presence of Amp? Can the authors comment?

In Figure 1C (Figure 1F in the revised manuscript) we present the ratio between the slope of reactions with or without LpoB (relative slope = slope LpoB / slope no activator). Hence, Figure 1C indicates that the presence of LpoB causes a larger change in reaction slopes in the presence of ampicillin than in its absence. This was the same for all pairs of PBP1B-activator pairs studied, except for AbPBP1B/AbLpoP in detergents. The slope changes produced by activators in the presence of ampicillin reflect the acceleration of the GTase, while the slope changes produced in the absence of antibiotic reflect the combined effect of regulators on GTase and TPase rates. As we cannot separate the TPase and GTase rates from the curves without ampicillin, it is difficult to know what exactly causes this difference. In the revised manuscript we discuss the challenge to separating the effect of an Lpo activator on the GTase rates from the effect on the TPase rates.

[Editors' note: further revisions were suggested prior to acceptance, as described below.]

Major comments:The major advantages of the assay are three-fold, (1) it is a real-time; (2) it is applicable to membrane-embedded enzymes, and (3) it can monitor both GTase and TPase. The revised manuscript has provided evidence for all the three with improved clarity, new control experiments, and better FRET efficiency calculations, but it still lacks the rigor in the quantitative (or semi-quantitative) aspect of the assay. Specifically, the authors need to provide evidence that the F_acceptor_/F_donor_ ratio indeed reflects proportionally to either just the GTase activity or a combined GTase and TPase activity. The authors did show qualitatively that this ratio is altered when the activator LpoB (or P) or drugs are used, but it is difficult to use this ratio to compare quantitatively (or semi-quantitatively) the rates of different reactions catalyzed by different enzymes, or the same enzyme but with different mutations and regulators. Being able to do so will greatly enhance the impact of the FRET assay and promote its wide adoption in the field.

We thank the reviewers for appreciating the advantages of the new assay. The control experiment shown in Figure 2AB and other experiments with inhibitors (Figures 1E; Figure 3C,E,G; Figure 3—figure supplement 7A, Figure 3—figure supplement 7B), demonstrate that the increase in F_acceptor_/F_donor_ ratio is proportional to the GTase or GTase-TPase rates. However, it is currently not possible to precisely quantify their respective contributions (see our responses below).

We agree with the reviewers that we cannot provide quantitative comparison of the catalytic rates of different enzyme based only on FRET increase. This is due to an inherent limitation of using (fluorescent) modified substrates in an enzymatic assay: different enzymes can potentially have different rates for utilising those substrates. We added a paragraph about the limitations of the assay to Discussion.

As for the effect of activators, we performed a new experiment, added the data to the manuscript (Figure 2—figure supplement 1) and present these in the section “Intra-chain versus Inter-chain FRET”. These data clarify the effect of activators on the final FRET signal and on the slopes of the reaction curves.

With the appropriate control reactions (β-lactams, using only labelled lipid II) it was possible to determine in every case whether the FRET signal follows more closely the GTase (intra-chain FRET) or the TPase (inter-chain FRET) activities.

In conclusion, we are convinced that our novel FRET assay can be used to measure PG synthesis activity of a wide variety of enzymes and that it offers a straightforward approach for the biochemical screening of inhibitors and paves the way for detailed mechanistic studies of PG synthases.

It appears that the measured F_acceptor_/F_donor_ ratio is not only dependent on the GTase and TPase activities of PBP1B, but some other effects that influence the fluorescence of the dyes. These other effects will make it difficult to use the FRET assay to quantitatively compare different reactions. For example, in Figure 3C, the reaction with Amp (VII) barely had any signal change in F_acceptor_/F_donor_, but the corresponding gel lane cleared showed polymerized, linear PG strands, suggesting that the polymerization reaction is not reflected in the fluorescence ratio measurement. The rise and decay of the +LpoB/Amp (III) reaction also look strange-this reaction had a similar PG ladder on the gel as reaction (-LpoB/Amp)VII, but the fluorescence ratio measurements are drastically different. Another example is the comparison between the three PBP1Bs: it appears that crosslinking activities have different contributions to the fluorescence ratio measurement: ampicillin completely abolished the fluorescence ratio signal for PBP1BEc but not for PBP1BAb. It is hard to imagine that these highly homologous enzymes would have very different mechanisms to synthesize PG.

We agree that the results with PBP1B in the membrane require careful controls and that the interpretation is more difficult in part due to the lower activity of membrane-reconstituted PBP1B^Ec^ in the absence of LpoB. Therefore, in new experiments we performed reactions of PBP1B^Ec^ in liposomes for 90 min (instead of 60 min), as for the reactions with PBP1B^Pa^ and PBP1B^Ab^ in liposomes. We think this new data shows more clearly the differences in activity between PBP1B^Ec^ in the membrane with and without LpoB, and we have updated Figure 3C accordingly.

The reviewers correctly observed that the polymerization reaction seen by SDS-PAGE is not always reflected in the fluorescence ratio measurement (FRET). This is true for several reasons:

i) We noticed that the image software automatically adjusts the contrast of files before exporting them, making weaker bands (e.g. bands for short strands) easier to visualize but misrepresenting their contribution to the total signal. We added a section to Materials and methods about fluorescent scanning of gels and their analysis, in which we mention the image processing. For quantification of gel signals in Figure 1 —figure supplement 5 we used the original image files without contrast adjustment. We have now uploaded the original image files for all gels presented as source data files.

ii) Regarding the examples discussed by the reviewers: The reactions with PBP1B^Ec^ liposomes with ampicillin without LpoB (-LpoB/+Amp, VII in previous Figure 3C, II in current Figure 3C) contain much more unused lipid II in their corresponding gel lanes than reactions in which LpoB was present (+LpoB/+Amp, III in previous Figure 3C, VI in current Figure 3C), consistent with a smaller signal change in F_acceptor_/F_donor_ in the corresponding reactions curves. We investigated the rise and decay of the signal in the +LpoB/+Amp reaction with new experiments (see below, point iv).

iii) As discussed in the section “Intra-chain FRET versus inter-chain FRET”, our data suggests that with LpoB, PBP1B produces PG chains with more densely packed fluorophores. This is consistent with the glycan chains produced in the presence of Ampicillin without LpoB showing less FRET than with LpoB.

iv) We were also puzzled by the decaying FRET signal for +LpoB/Amp condition, which we previously had tested for a single preparation of liposomes. Thus, we repeated the assay at the same conditions (LpoB/Amp) several times with two new liposome preparations and using ampicillin from two different suppliers. We did not consistently observe this decay in FRET during the reactions and instead observed that the reaction at these conditions had two possible outcomes even with the same liposome preparation and ampicillin supplier: (i) (most often): no FRET-decay during reactions and long PG chains produced; (ii) (less often) FRET-decay and short chains at the end of reactions. We cannot currently explain why reactions of PBP1B^Ec^ liposomes with LpoB/Amp (and only these reactions) can have the two outcomes. It appears that the decay in the FRET signal did not happen because of environmental effect on the probes, but because of different enzyme activity. We have updated the data on Figure 3C and added a comparison of the different possible outcomes of the assay in this particular condition on Figure 3—figure supplement 2A. We comment on these results in section “FRET assay to monitor PG synthesis in liposomes”.

We disagree with the reviewers’ statement that "It is hard to imagine that these highly homologous enzymes would have very different mechanisms to synthesize PG.". Actually, the similarity between different PBP1B homologues is not so high. Sequence comparison shows 33% identity and 45% similarity between PBP1B^Ab^ and PBP1B^Ec^, and 41% identity and 52% similarity between PBP1B^Pa^ and PBP1B^Ec^ and, importantly, their activators have different structures. Such differences could be the reason why, for example, ∆*lpoP* ∆PBP1A is viable in *P. aeruginosa* (Greene et al. 2018, PNAS 115(12):3150-3155), while ∆*lpoB* ∆PBP1A is lethal in *E. coli* (Paradis-Bleau et al., 2010, Cell 143(7):1110-20.), indicating that PBP1B^Pa^ has higher residual activity in the cell in the absence of activator.

A likely explanation for these different fluorescence signals in these reactions is probably what the authors suggested, that some spectroscopic properties of the dyes changed due to the incorporation and/or local environment. Another possibility is that these enzymes of different species may handle the dye-labeled LipidII differentially, one PBP1B may have more difficulty in accepting the dye-labeled LipidII than the other PBP1B. Yet another possibility could also be that the stimulation effect of LpoB is simply that now more dye-labeled LipidII v.s. unlabeled Lipid II can be incorporated, but not necessarily the rate at which Lipid II is incorporated.

We appreciate the discussion of these possibilities by the reviewers and believe the new controls added to the paper clarify most of their hypothesises:

1) Changes in spectroscopic properties of the dyes due to incorporation to PG or local environment effects: the end point controls with donor-only and acceptor-only as fluorescent lipid II versions show that FRET occurs. In addition, the new real-time controls with donor-only or acceptor-only as fluorescent lipid II version in detergents or liposomes (Figure 1—figure supplement 4 and Figure 3—figure supplement 4) demonstrate that changes in donor or acceptor fluorescence are minimal unless there is PG synthesis and both fluorophores are present in the reaction. Finally, as discussed above, we demonstrate that the FRET decay observed in some reactions with PBP1B^Ec^ liposomes and LpoB/Amp reflects a change in the type of glycan chains produced (Figure 3—figure supplement 2A).

2) Different enzymes having different reactivity towards labelled lipid II: we always assumed that this can be the case. This is an inherent limitation of any assay that uses fluorescent labelled substrates and is now discussed in the text (Discussion).

3) LpoB might affect the rate of incorporation of dye-labelled lipid II versus incorporation of non-labelled lipid II: It is known that LpoB stimulates the GTase rate with non-fluorescent lipid II (e.g. Figure 4 in Lupoli et al., 2014 J Am Chem Soc. 136:52-55). The new data in Figure 2—figure supplement 1 commented in the section “Intra-chains versus Inter-chain FRET” show that LpoB does not increase the proportion of dye-labelled lipid II that gets incorporated into PG and suggests that the faster FRET increase in the presence of LpoB is not only caused by faster GTase rate but also reflects a more dense packing of the fluorophores in the final PG product. Presumably, LpoB allows PBP1B^Ec^ to incorporate labelled substrate molecules more densely along the glycan chain.

It is understandable that these in vitro experiments are extremely challenging, but fluorescence measurements are sensitive and prone to artifacts, and the authors should perform control experiments to make sure that the measured signal indeed reflects the desired enzymatic activities. In particular, in calculating the fluorescence ratio, the donor-only and acceptor-only controls need to be run simultaneously to determine how much fluorescence change is due to the local environment change, and how much is due to FRET. The authors have performed these control experiments but only used them for the end-point measurements. These controls should also be performed for the real-time measurements (with LpoB, with or without Amp), which could help to explain some strange changes in the measured fluorescence ratio (such as that in Figure 3C).

As requested by the reviewers and explained above, we incorporated these real-time controls for detergents and liposomes reactions in Figure 1—figure supplement 4 and Figure 3—figure supplement 4, respectively. In both cases, the changes in donor or acceptor fluorescence are minimal unless they are both present in the reaction with active enzyme.

The goal, hopefully, is that one can confidently interpret the ~ 25-fold higher initial slope of PBP1BEc with LpoB in the presence of ampicillin as a 25-fold (or proportionally) higher GTase activity of PBP1BEc stimulated by LpoB (such as that in Figure 3D). Similarly, one can use the smaller fold-increase of the initial slopes in the cases of PBP1BAb and PBP1BPa in the presence of their cognate activators to draw the conclusion that these activators do not stimulate these PBP1Bs to the same extent as that of PBP1BEc. If the fold enhancement of the initial slopes by LpoB in the presence and absence of ampicillin are similar to each other, one perhaps can say that here LpoB only stimulates the GTase activity, and so on and so forth.

We fully agree with the reviewer that an ideal assay would precisely quantify the GTase and TPase rates in detergents and the membrane. However, this was not our goal when we set up the FRET assay and, indeed, by its nature, the FRET assay cannot provide absolutely quantitative data for reaction rates for several reasons:

1) FRET depends on the average distances and orientation of the fluorophore molecules on the growing glycan chains and the resulting PG. We cannot know the distribution and orientation of the fluorophores on the growing glycan chain and whether these parameters remain constant during the synthesis of a glycan chain and there is currently no method to determine these parameters.

2) Different PBPs may produce PG chains with different distribution/density of fluorophores, depending on their affinities to the different lipid II variants.

3) There is currently no methodology available to validate our data, as there is no other published assay that allows quantification of absolute GTase rates.

We discuss these limitations in a new section “Limitation of the FRET assay”. The new real-time control with donor-only or acceptor-only as labelled substrates (Figure 1—figure supplement 4, Figure 3—figure supplement 4) don't support the idea that the fluorophores are highly sensitive to the environment and reinforce our conclusion that FI_acceptor_/FI_donor_ ratio is specific for FRET. The new data in Figure 2—figure supplement 1 clarify what is the cause of the changes in FRET of the produced PG when LpoB is added, and help to interpret FRET curves with or without activators.

Our motivation to develop this assay was not to quantify the GTase rates for a detailed mechanistic comparison between different enzymes. Instead, we aimed to develop the first method to monitor PG synthesis continuously in a biomimetic membrane. Assays with fluorescently-labelled substrates do not report directly on the reaction rates of the enzymes for the unlabelled substrate, complicating precise comparisons between different enzymes (or combinations of enzyme and activator). However, these fluorescent substrates are the key to the specific and sensitive real time detection of PG synthesis in the membrane. In addition, as the assay does not require a complicated setup, it has the potential to be widely used in basic research and for screening for inhibitors. Hence, we believe that the assay will be widely used by the PG community despite its semi-quantitative nature.